# Association of human breast cancer CD44⁻/CD24⁻ cells with delayed distant metastasis

Xinbo Qiao[1†], Yixiao Zhang[1,2†], Lisha Sun[1†], Qingtian Ma[1†], Jie Yang[1], Liping Ai[1], Jinqi Xue[1], Guanglei Chen[1], Hao Zhang[1,3], Ce Ji[1,4], Xi Gu[1], Haixin Lei[5], Yongliang Yang[6], Caigang Liu[1]*

[1]Department of Oncology, Shengjing Hospital, China Medical University, Shenyang, China; [2]Dapartment of Urology, Shengjing Hospital, China Medical University, Shenyang, China; [3]Department of Breast Surgery, Liaoning Cancer Hospital and Institute, Cancer Hospital of China Medical University, Shenyang, China; [4]Department of General Surgery, Shengjing Hospital, China Medical University, Shenyang, China; [5]Institute of Cancer Stem Cell, Cancer Center, Dalian Medical University, Dalian, China; [6]Center for Molecular Medicine, School of Life Science and Biotechnology, Dalian University of Technology, Dalian, China

*For correspondence:
angel-s205@163.com

[†]These authors contributed equally to this work

**Abstract** Tumor metastasis remains the main cause of breast cancer-related deaths, especially delayed breast cancer distant metastasis. The current study assessed the frequency of CD44⁻/CD24⁻ breast cancer cells in 576 tissue specimens for associations with clinicopathological features and metastasis and investigated the underlying molecular mechanisms. The results indicated that higher frequency (≥19.5%) of CD44⁻/CD24⁻ cells was associated with delayed postoperative breast cancer metastasis. Furthermore, CD44⁻/CD24⁻triple negative breast cancer (TNBC) cells spontaneously converted into CD44⁺/CD24⁻cancer stem cells (CSCs) with properties similar to CD44⁺/CD24⁻CSCs from primary human breast cancer cells and parental TNBC cells in terms of stemness marker expression, self-renewal, differentiation, tumorigenicity, and lung metastasis in vitro and *in NOD/SCID mice*. RNA sequencing identified several differentially expressed genes (DEGs) in newly converted CSCs and *RHBDL2*, one of the DEGs, expression was upregulated. More importantly, *RHBDL2* silencing inhibited the YAP1/USP31/NF-κB signaling and attenuated spontaneous CD44⁻/CD24⁻ cell conversion into CSCs and their mammosphere formation. These findings suggest that the frequency of CD44⁻/CD24⁻ tumor cells and *RHBDL2* may be valuable for prognosis of delayed breast cancer metastasis, particularly for TNBC.

## Introduction

Breast cancer is the most prevalent malignancy in women and its incidence is increasing globally, especially in developed countries (*Bianchini et al., 2016*; *Siegel et al., 2020*). Different treatment strategies, such as surgical resection, hormone therapy, targeted therapy, radiation therapy, and chemotherapy, have greatly improved survival of breast cancer patients (*Bray et al., 2018*; *Burstein et al., 2014*; *Early Breast Cancer Trialists' Collaborative G, 2015*; *Khan et al., 2012*; *Saini et al., 2012*). To date, clinical strategies for breast cancer treatment remain suboptimal. Although continuous treatment with tamoxifen for 10 years has reduced cancer recurrence and mortality of patients with luminal A breast cancer, a significant proportion of patients may be overtreated (*Bianchini et al., 2016*; *O'Conor et al., 2018*). Hence, it is urgently needed to discover novel biomarkers to predict treatment effectiveness and to improve treatment success and prognosis of breast cancer patients. Furthermore, the available therapies can also lead to a considerable number

DOI: https://doi.org/10.7554/eLife.65418

of patients at a risk to develop a delayed breast cancer metastasis, which occurs even 20–40 years after breast cancer diagnosis (*Sharma, 2018*). Notably, breast cancer metastasis occurring 5–8 years after initial surgical resection has become a significant cause of cancer relapse, progression, and poor survival in patients (*Nishimura et al., 2013*); thus, further researches on its molecular mechanisms and gene alterations may help identify novel biomarkers and targets for development of therapeutic strategies to effectively control breast cancer metastasis and progression.

Indeed, tumor metastasis is a multistep process, during which, cancer cells escape from the primary site, migration into a neighboring or distant tissues, extravasation, survival, and colonization, leading to the formation of new tumor nodules at a secondary site (*Drabsch and ten Dijke, 2011*; *Klein, 2008*; *Scott et al., 2012*; *Syn et al., 2016*). The rate-limiting step of cancer metastasis is cancer cell colonization and proliferation at the secondary site, because the initial metastatic cancer nodule usually lacks an efficient vasculature to provide sufficient nutrients to support cancer cell growth. Thus, the newly arrived tumor cells may grow to a certain size in the new and harsh microenvironment and undergo growth arrest in that organ. However, once they regain their proliferative ability, delayed metastasis will occur (*Langley and Fidler, 2007*). Molecularly, CD44$^+$/CD24$^-$ breast cancer cells from primary breast tumors are associated with distant metastasis (*Abraham et al., 2005*), and these cells display potent motility and invasiveness (*Liu et al., 2010*), similar to chemoresistance cancer stem cells (CSCs) (*Velasco-Velázquez et al., 2011*). Previous studies have shown that CD44$^+$/CD24$^-$ breast CSCs may be a dominant factor for the relapse of triple negative breast cancer (TNBC), due to their potent self-renewal and differentiation capacities (*Geng et al., 2014*; *Wang et al., 2014*). Indeed, injection with about 50 breast CSCs can induce a solid tumor mass in immunocompromised mice (*Chaffer et al., 2011*; *Iliopoulos et al., 2011*). Thus, the number of breast CSCs in the secondary site may affect the efficient formation of early metastatic nodules, and breast CSCs are commonly prone to be resistant to chemotherapy (*De Angelis et al., 2019*). Moreover, previous studies have reported that the differentiated cancer cells can spontaneously convert into CSCs to renew the CSC pool in breast cancer, pancreatic cancer and sarcomas, resulting in chemoresistance (*Gruber et al., 2016*; *Kim et al., 2015*; *Ye et al., 2018*). Thus, the dormant CD44$^-$/CD24$^-$ breast cancer cells that have previously been colonized in the metastatic site may be able to spontaneously convert into CD44$^+$/CD24$^-$ breast CSCs to regain their potent proliferative ability and drug resistance, resulting in delayed breast cancer metastasis. Accordingly, it is reasonable to hypothesize that the frequency of CD44$^-$/CD24$^-$ cells in human tumor specimens may be useful in the prediction of delayed breast cancer metastasis.

The current study aimed to explore the molecular mechanisms by which CD44$^-$/CD24$^-$ cell conversion into CSC promotes delayed breast cancer metastasis. First, a retrospective analysis of CD44$^-$/CD24$^-$ breast cancer cells in tissue specimens from patients enrolled from three academic medical centers was performed to investigate the potential associations between the frequency of CD44$^-$/CD24$^-$ cells and postoperative tumor metastasis. Next, the spontaneous CD44$^-$/CD24$^-$ cell conversion into CD44$^+$/CD24$^-$ CSCs was tested and the biological functions of the converted CSCs were analyzed in vitro and in vivo. The results may provide novel insights into the role of CD44$^-$/CD24$^-$ tumor cells in delayed breast cancer metastasis and into the potential use of CD44$^-$/CD24$^-$ cells as a biomarker to predict survival and metastasis in breast cancer patients. These findings also suggest that *RHBDL2* may be a novel therapeutic target for the future studies.

## Materials and methods

### Patients and tissue specimens

Paraffin-embedded surgical tissue samples were collected from 576 breast cancer patients, who underwent breast cancer surgery between June 2005 and April 2013 in China Medical University Affiliated Hospital (Shenyang, Liaoning, China), Liaoning Cancer Hospital (Shenyang, Liaoning, China), and Dalian Municipal Central Hospital Affiliated to Dalian Medical University (Dalian, Liaoning, China). The patients were diagnosed with invasive breast cancer histologically, according to the World Health Organization (WHO) breast cancer classifications, 4$^{th}$ edition (*Tan et al., 2015*) and classified, according to breast cancer TNM staging (*Li et al., 2012*). The demographic, clinicopathological and follow-up data of all patients were collected from their medical records or via telephone interview. The inclusion criteria were: surgical treatment for breast cancer; complete information

regarding clinicopathological characteristics; and complete follow-up data. Disease-free survival (DFS) was defined as the time from the date of surgery to the date of distant metastasis, while overall survival (OS) was defined as the time from the date of surgery to the date of death. The current study was approved by the Ethics Committee of all three hospital review boards (Project identification code: 2018PS304K, date on 03/05/2018), and each participant signed an informed consent form before being included in the study.

## Immunofluorescence staining

Paraffin-embedded tissue blocks of 576 breast cancer patients were prepared to construct a tissue microarray (*Edge and Compton, 2010*). The levels of CD44 and CD24 expression in the tissue microarray sections were assayed using a double immunofluorescence staining. Briefly, individual sections (4 μm) on the tissue microarray glass slides that had been pre-coated with (3-aminopropyl) triethoxysilane solution were deparaffinized, rehydrated, and subjected to regular antigen retrieval. The sections were blocked with 10% fetal bovine serum (FBS, Cellmax, Lanzhou, China) at room temperature for 1 h and incubated with a mouse anti-human CD44 monoclonal antibody (Cat. #3570; Cell Signaling Technology, Danvers, MA, USA) and a rabbit anti-human CD24 antibody (Cat. #ab202073; Abcam, Cambridge, MA, USA) at 4°C overnight. After being washed with phosphate-buffered saline (PBS), the sections were incubated with Alexa Fluor 647-labeled rabbit goat anti-mouse IgG and Alexa Fluor 488-labeled goat anti-rabbit IgG, followed by nuclear staining with 4',6-diamidino-2-phenylindole (DAPI). The fluorescence signals in the immunostained tissue sections were photoimaged under a fluorescence microscope (E800, Nikon, Tokyo, Japan) with the NIS-Elements F3.0 (Nikon) and analyzed using ImageJ software (National Institute of Heath, Bethesda, MD, USA). The percentages of $CD44^+/CD24^-$ CSCs and $CD44^-/CD24^-$ cells in 2000 tumor cells from at least three sections were calculated in a blinded manner. The tumor cells were identified, based on hematoxylin and eosin (H&E) staining in their consecutive sections (see below).

The consecutive tissue sections were deparaffinized, rehydrated and routine-stained with H&E solution. They were photoimaged and the numbers of tumor cells in each section were quantified by two pathologists in a blinded manner.

## Cell lines and culture

Human TNBC MDA-MB-231 and MDA-MB-468 cells were obtained from American Type Culture Collection (ATCC; Manassas, VA, USA). The identity of these cell lines was confirmed by STR and the cells were tested negative for mycoplasma contamination throughout the experimental period. The cells were maintained in Leibovitz's L15 medium (Thermo Fisher, Carlsbad, CA, USA) supplemented with 10% fetal bovine serum (FBS), 100 U/ml penicillin, and 100 μg/ml streptomycin (as the complete L15 medium) in a humidified incubator without addition of $CO_2$ at 37°C.

The sorted $CD44^-/CD24^-$ cells (see below) were cultured in the complete L15 medium or the stem cell (SC) medium (10% human MammoCult Proliferation Supplements in MammoCult Basal Medium, Stem Cell Technologies, San Diego, CA, USA) for 7 days. The cells were subjected to different assays (see below).

## Cell transfection

The purified $CD44^-/CD24^-$ MDA-MB-231 cells ($5 \times 10^5$ cells/well) were cultured in 6-well plates overnight and transfected with the *RHBDL2*-specific siRNA (5'-CAUACUUGGAGAGAGAGCUAATT-3'), or negative control scramble siRNA (5'-UUCUCCGAACGUGUCACGUTT-3') from GenePharma (Shanghai, China) using Mission siRNA transfection reagents (Sigma-Aldrich, St. Louis, MO, USA) for 48 h. The efficacy of specific gene silencing was evaluated by Western blot analysis.

## Flow cytometry

Parental MDA-MB-231 (wild-type, WT), *RHBDL2*-silenced MDA-MB-231, MDA-MB-468/Ctrl, and MDA-MB-468/si cells were stained with fluorescein isothiocyanate (FITC)-conjugated anti-CD24 (Cat. #311104; BioLegend, San Diego, CA, USA) and PE-conjugated anti-CD44 (Cat. #338808; BioLegend) antibodies. Control cells were stained with an isotype control, FITC-anti-CD24 or PE-anti-CD44 alone. Subsequently, the percentages of $CD44^-/CD24^-$, $CD44^-/CD24^+$, $CD44^+/CD24^+$, and $CD44^+/CD24^-$ cells were analyzed by flow cytometry in a FACS Aria III flow cytometer (BD Biosciences, San

Jose, CA, USA). The same protocol was used for analyze the CD44-/CD24- cells after culture for 7 days and freshly prepared xenograft tumor cells.

## Western blotting

The different groups of cells were lyzed in radioimmunoprecipitation (RIPA) lysis buffer containing phenylmethane sulfonyl fluoride (PMSF), protease and phosphatase inhibitors and centrifuged. The cell lysate supernatants were collected, and the total protein concentrations were measured using a Pierce BCA Protein Assay Kit (Thermo-Fisher), according to the manufacturer's instructions. Next, the cell lysate samples (50 µg/lane) were separated by sodium dodecyl sulfate-polyacrylamide gel electrophoresis (SDS-PAGE) on 12% gels and transferred onto polyvinylidene difluoride (PVDF) membranes (Millipore, Billerica, MA, USA). The membranes were blocked in 5% nonfat dry milk in Tris-based saline-Tween 20 (TBS-T) and probed overnight at 4°C with various primary antibodies (*Supplementary file 4*). The bound antibodies were detected with horseradish peroxidase (HRP)-conjugated secondary antibodies (1:10000 dilution; Cat. #ZDR-5306, ZDR-5307, or ZSGB-BIO, Beijing, China), and the immunoblotting signals were visualized using enhanced chemiluminescence reagents (Cat. #34076; Thermo-Fisher, Waltham, MA, USA) on a chemiluminescence instrument C300 (Azure, Dublin, CA, USA). The relative levels of individual targeted proteins to control glyceraldehyde-3-phosphate dehydrogenase (GAPDH) were determined by densitometric analysis using ImageJ software.

## RNA sequencing (RNA-seq) analysis of single cells

Human *OCT4* promoter region (1-2012 bps) was amplified by polymerase chain reaction (PCR). The generated DNA fragment, together a DNA fragment for encoding the *enhanced green fluorescent protein (EGFP)*, was cloned into the plasmid of pGL3-basic to generate a pGL3-OCT4-EGFP plasmid for the *OCT4* promoter–controlled EGFP expression. The plasmid was sequenced. Next, the purified CD44-/CD24- MDA-MB-231 cells were transfected with pGL3-OCT4-EGFP using Lipofectamine 3000 (Invitrogen, Carlsbad, CA, USA) and 6 h later, the cells were cultured into microraft QuAscount assay plates (Teacon, Maennedorf, Switzerland) in a single cell manner for 1, 3 or 5 days, according to the manufacturer's protocol. The EGFP- cells at 24 h post culture and EGFP+ cells at 72 and 120 h post culture (three cells per time point) were captured separately and subjected to RNA-seq analysis at a single cell level to identify differentially expressed genes (DEGs). In brief, total RNA was isolated from individual single-cell samples, and their mRNA was enriched using the oligo-dT microbeads and fragmented to 300-500 nucleosides, followed by reversely transcribed into cDNA. The cDNA samples were amplified by PCR to generate cDNA libraries, which were sequenced in an Illumina HiSeq (Illumina, San Diego, CA, USA). The high-quality reads were aligned to the mouse reference genome (GRCm38) using the Bowtie2 v2.4.2 (Baltimore, MD, USA), and the expression levels of individual genes were normalized to the fragments per kilobase of the exon model per million mapped reads from RNA-seq by expectation maximization. The DEGs were considered if the gene expression level had a fold change of >2 and had an adjusted p-value of <0.05 between two time points.

## Bioinformatics

The DEGs were further analyzed by gene ontology (GO) using the online tool AmiGO (http://www.geneontology.org) and Database for Annotation, Visualization and Integrated Discovery (*Burstein* et al.) software. The potential pathways the DEGs involved were analyzed using the Kyoto Encyclopedia of Genes and Genomes (KEGG; http://www.genome.jp/kegg) annotations.

## A mouse model of xenograft tumor and lung metastasis assays

The experimental protocol was approved by the Animal Research and Care Committee of China Medical University (Shenyang, China; Project identification code: 2018PS312K, date on 03/05/2018), according to the Guidelines of the Care and Use of Laboratory Animals issued by the Chinese Council on Animal Research. Female BALB/c nude mice (6 weeks old) were obtained from Human Silaikejingda Laboratory Animals (Changsha, China) and housed in a specific pathogen-free facility with free access to autoclaved food and water. The purified CSCs from parental MDA-MB-231 and CD44-/CD24- converted MDA-MB-231 CSCs ($2 \times 10^3$ cells/mouse) were injected subcutaneously into individual BALB/c nude mice. Their tumor growth and body weights were monitored up to 21

days post-cancer cell inoculation. At the end of the experiment, subcutaneous tumors were dissected and weighed. In addition, some tumor xenografts were dissected from each group of mice at 7 and 21 days post inoculation and digested to prepare single-cell suspensions for staining with FITC-anti-CD24, PE-anti-CD44, or isotype controls for flow cytometric sorting of CD44$^+$/CD24$^-$, CD44$^-$/CD24$^-$, CD44$^-$/CD24$^+$, and CD44$^+$/CD24$^+$ cells.

Moreover, the sorted CD44$^+$/CD24$^-$ CSCs (1 × 10$^3$ cells/mouse) from parent MDA-MB-231 cells and CD44$^-$/CD24$^-$ converted MDA-MB-231 cells were injected into the tail vein of NOD/SCID mice and on 21 days post inoculation, the mice were euthanized and their lungs were dissected and weighed (including bronchi). Additionally, the dissected lung tissue sections were stained with H and E and photographed (n=5–8 per group). The sizes of metastatic breast cancer nodules in individual mice were measured in a blinded manner.

## Statistical analysis

The data are expressed as the mean ± standard deviation (SD) from at least three separate experiments, and the difference between the groups was analyzed by Chi-Square test, Student's t tests, or Mann–Whitney U test where applicable. The DFS and OS of each group of patients were estimated by the Kaplan-Meier method and analyzed by the log-rank test. All statistical analyses were performed using SPSS 23.0 (SPSS, Inc, Chicago, IL, USA). A p-value of $\leq$ 0.05 was considered statistically significant.

## Results

### Association of a higher frequency of CD44$^-$/CD24$^-$ tumor cells with delayed distant metastasis in human breast cancer patients

CD44$^+$/CD24$^-$ breast CSCs are crucial for the prognosis of breast cancer (*Kaverina et al., 2017*; *Mylona et al., 2008*), but the potential prognostic values of CD44$^-$/CD24$^-$ breast cancer cells are rarely studied. Based on H and E staining for identification of tumor cells, immunofluorescence was used to quantify the frequency of CD44$^-$/CD24$^-$, CD44$^+$/CD24$^-$, CD44$^-$/CD24$^+$, and CD44$^+$/CD24$^+$ tumor cells in 576 breast cancer tissue specimens, including a training group (n = 355) and testing group (n = 221) using anti-CD44 and anti-CD24 antibodies. Their demographic and clinical characteristics are summarized in *Supplementary file 1* and representatively histological and immunofluorescent images are shown in *Figure 1—figure supplement 1*. The arrows are separately indicating CD44$^+$/CD24$^-$ cells and CD44$^-$/CD24$^-$ cells in *Figure 1—figure supplement 1A and B*. There were varying percentages of CD44$^-$/CD24$^-$, CD44$^+$/CD24$^-$, CD44$^-$/CD24$^+$, and CD44$^+$/CD24$^+$ tumor cells in *Figure 1—figure supplement 2A*. After stratification of the training group of patients, based on their distant metastasis, the frequency of CD44$^-$/CD24$^-$ cells in breast cancer tissues of patients with postoperative distant metastasis was significantly higher than those without distant metastasis (p<0.0001; *Figure 1A*). The average frequency of CD44$^-$/CD24$^-$ cancer cells in all samples was 19.7% with a median value of 19.5%. The ROC analysis revealed a cut-off value of 19.5% with a sensitivity of 70.5%, specificity of 71.2% and Youden index of 0.471 (*Figure 1B*; *Supplementary file 2*). Accordingly, the patients in the training set were stratified into two subgroups with high or low frequency of CD44$^-$/CD24$^-$ cells. In the training set of patients, the metastasis rate in the patients with a high frequency (>19.5%) of CD44$^-$/CD24$^-$ breast cancer cells was 1.97-fold higher than those with a low frequency (<19.5%) of CD44$^-$/CD24$^-$ tumor cells. Analysis of different molecular subtypes of breast cancers indicated that the metastatic rates in the patients with a high frequency of CD44$^-$/CD24$^-$ breast cancer cells were significantly higher than those with a low frequency of this type of cells (i.e. 40.85% vs. 6.82% for luminal; 47.06% vs. 13.64% for HER-2+; 63.79% vs. 22.22% for TNBC with a high vs. low frequency of CD44$^-$/CD24$^-$ tumor cells; *Figure 1C*). Univariate and multivariate analyses indicated that high frequency of CD44$^-$/CD24$^-$ cancer cells or CSCs, positive lymph node metastasis, higher N stage, and TNBC were independent risk factors for poor DFS (*Supplementary file 3*).

Furthermore, analysis of metastatic dynamics revealed that patients with a higher frequency ($\geq$2%) of CD44$^+$/CD24$^-$ CSCs usually developed distant metastasis earlier than those with < 2% of CD44$^+$/CD24$^-$ and >19.5% of CD44$^-$/CD24$^-$ tumor cells (the C1 group, *Figure 1D*). Similarly, the DFS of patients with a high frequency of CD44$^-$/CD24$^-$ tumor cells or CD44$^+$/CD24$^-$ CSCs was

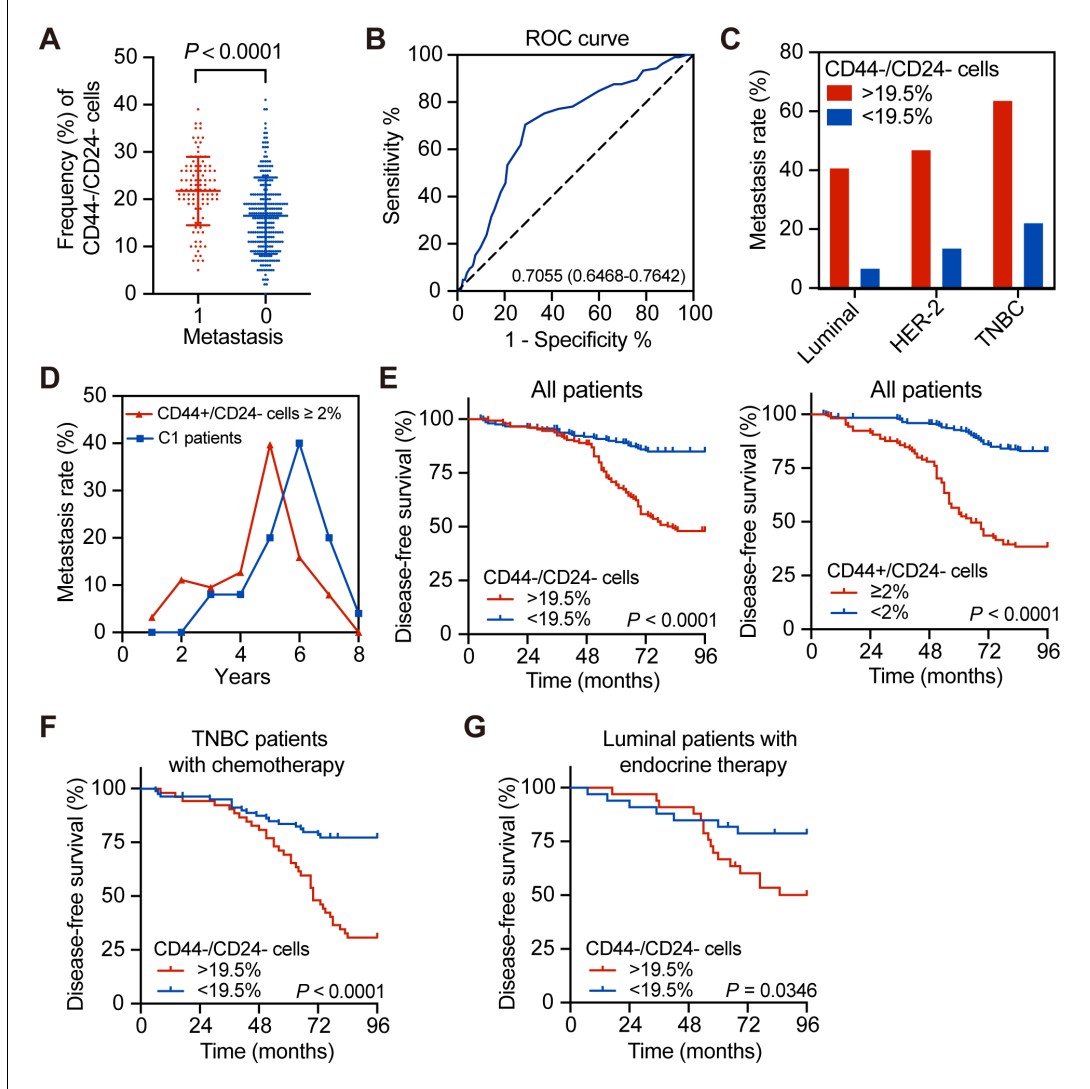

**Figure 1.** Association of a high frequency of CD44⁻/CD24⁻ cells in breast cancer tissues with delayed tumor distant metastasis. (**A**) Immunofluorescent analysis of the percentages of CD44⁻/CD24⁻ cells in tissue samples from breast cancer patients with or without tumor metastasis. The *P*-value was determined by Student's *t*-test. N = 105 and 250 for with and without metastasis, respectively. (**B**) ROC analysis of the sensitivity and specificity of 19.5% of CD44⁻/CD24⁻ cells as a cutoff value for evaluating delayed distant metastasis. (**C**) Metastasis rates in patients with different molecular subtypes of breast cancer after they were stratified into high (≥19.5%) or low frequency of CD44⁻/CD24⁻ cells. (**D**) Metastasis rates in patients with ≥2% CD44⁺/CD24⁻ cells (n = 105) vs. C1 patients with <2% CD44⁺/CD24⁻ cells and ≥19.5% CD44⁻/CD24⁻ cells (n = 69). (**E**) Kaplan–Meier analysis of the DFS of breast cancer patients after they were stratified, based on the indicated measure. (**F**) Kaplan–Meier analysis of the DFS of TNBC patients with standard chemotherapy after they were stratified into ≥19.5% CD44⁻/CD24⁻ cells (n = 52); <19.5% CD44⁻/CD24⁻ cells (n = 80). (**G**) Kaplan–Meier analysis of the DFS of luminal breast cancer patients with standard endocrine therapy after they were stratified into ≥19.5% CD44⁻/CD24⁻ cells (n = 24) vs. <19.5% CD44⁻/CD24⁻ cells (n = 23).

The online version of this article includes the following source data and figure supplement(s) for figure 1:

**Source data 1.** Source data for *Figure 1*.

**Figure supplement 1.** Immunofluorescent analysis of CD44⁺/CD24⁻ and CD44⁻/CD24⁻ cancer cells in human breast cancer specimens.

**Figure supplement 1—source data 1.** H&E staining and immunofluorescent images in *Figure 1—figure supplement 1*.

**Figure supplement 2.** A higher frequency of CD44⁻/CD24⁻ cancer cells, like CSCs, is associated with delayed distant metastasis and a high frequency of CSCs is related to worse.

**Figure supplement 2—source data 1.** Kaplan–Meier analysis data of *Figure 1—figure supplement 2*.

**Figure supplement 3.** A high frequency of CD44⁻/CD24⁻ cells is associated with worse DFS of breast cancer patients.

**Figure supplement 3—source data 1.** Kaplan–Meier analysis data of *Figure 1—figure supplement 3*.

significantly shorter than those with a low frequency of corresponding cells (*Figure 1E*). The similar patterns of DFS were observed in three subtypes of breast cancer patients (*Figure 1—figure supplement 2D* and *Figure 1—figure supplement 3A*). The metastatic dynamics also indicated that patients with a high frequency of CSCs had a metastatic peak near 5 years after surgical resection of the tumor (*Figure 1—figure supplement 2C*). Some patients with a higher frequency of CD44$^-$/CD24$^-$ tumor cells and CSCs had the highest rate of postoperative metastasis in this population. However, to exclude the effects of CSCs, patients in the C1 group also had a higher risk to develop delayed distant metastasis during 5–7 years post tumor resection than those with <2% of CD44$^+$/CD24$^-$ CSCs and < 19.5% of CD44$^-$/CD24$^-$ tumor cells (the C0 group, *Figure 1E*). Because postoperative therapies also affect the DFS, we further analyzed the frequency of CD44$^-$/CD24$^-$ cells in TNBC patients with chemotherapy and luminal breast cancer patients with hormone therapy. As shown in *Figure 1F and G*, TNBC and luminal breast cancer patients with a higher frequency of CD44$^-$/CD24$^-$ cells had a worse DFS, compared than those with a lower frequency of CD44$^-$/CD24$^-$ cells following standard therapies. Hence, a higher frequency of CD44$^-$/CD24$^-$ breast cancer cells, like higher frequency of CD44$^+$/CD24$^-$ CSCs, was associated significantly with a shorter DFS in the training set of patients regardless of standard therapies.

## The frequency of CD44$^-$/CD24$^-$ cells predicts the delayed distant metastasis in breast cancer patients following standard postoperative treatment

To validate the importance of CD44$^-$/CD24$^-$ cell frequency in delayed metastasis, the testing set of patients was analyzed and the metastatic rates of patients with a higher frequency of CD44$^-$/CD24$^-$ cells were higher than those with a lower frequency of CD44$^-$/CD24$^-$ cells in the tested molecular subtypes of breast cancers (22.73% vs. 8.89% for luminal; 50% vs. 18.42% for HER-2+; 41.67% vs. 15% for TNBC with high vs. low CD44$^-$/CD24$^-$ cells; *Figure 2A*). The DFS and OS of patients with a higher frequency of CD44$^-$/CD24$^-$ cells were significantly shorter than those with a lower frequency of CD44$^-$/CD24$^-$ cell cells in the testing set (*Figure 2B,C*). Similarly, the DFS and OS of patients with a higher frequency of CD44$^-$/CD24$^-$ cells were significantly shorter than those with a lower frequency of them, regardless of their molecular subtypes (*Figure 2—figure supplement 1A,B*). The metastatic dynamics revealed that while higher and earlier metastatic rates were observed in the testing set of patients, particularly for those with a high frequency of CD44$^+$/CD24$^-$ CSCs and those with < 2% of CD44$^+$/CD24$^-$ CSCs and >19.5% of CD44$^-$/CD24$^-$ tumor cells in the (C1 group) developed delayed distant metastasis, which peaked between 4 and 5 years post tumor resection (*Figure 2D*).

Furthermore, following standard chemotherapy or endocrine therapy, the patients with a higher frequency of CD44$^-$/CD24$^-$ tumor cells had a significantly shorter DFS than their corresponding patients with a lower frequency of CD44$^-$/CD24$^-$ tumor cells (*Figure 2E,F*). Similar results were achieved in the training set of patients following chemotherapy and the patients with a higher frequency of CD44$^-$/CD24$^-$ cells had a higher risk to develop distant metastasis beginning at 4 years post tumor resection (*Figure 1—figure supplement 3B*). Moreover, patients in the C1 group, with a low frequency of CSCs and higher frequency of CD44$^-$/CD24$^-$ tumor cells, also had a worse DFS than those with a lower frequency of CD44$^-$/CD24$^-$ tumor cells in the C0 group (*Figure 1—figure supplement 3C*). A similar pattern of DFS was observed in TNBC patients and those with a higher frequency of CD44$^-$/CD24$^-$ cells had a high risk to develop progressive metastasis at 4 years post tumor resection regardless of chemotherapy (*Figure 2E*). In this regard, the frequency of CD44$^-$/CD24$^-$ cells in tumor tissues could be a valuable predictor of standard therapeutic response in breast cancer patients. Similar data were observed in the testing group of patients (*Figure 2E* and *Figure 2—figure supplement 1C,D*). Thus, a higher frequency of CD44$^-$/CD24$^-$ tumor cells was associated with worse survival of breast cancer patients following standard therapies.

## Spontaneous conversion of CD44$^-$/CD24$^-$ TNBC cells into CD44$^+$/CD24$^-$ CSCs in vitro and in vivo

A previous study has shown that the differentiated breast cancer cells can spontaneously convert into CSCs to renew the CSC pool, although it remains unclear whether the CSCs derived from the differentiated breast cancer cells have the same biological behaviors as the original CSCs (*Najafi et al., 2019*). Because breast CSCs are crucial for the metastasis of breast cancer, whether

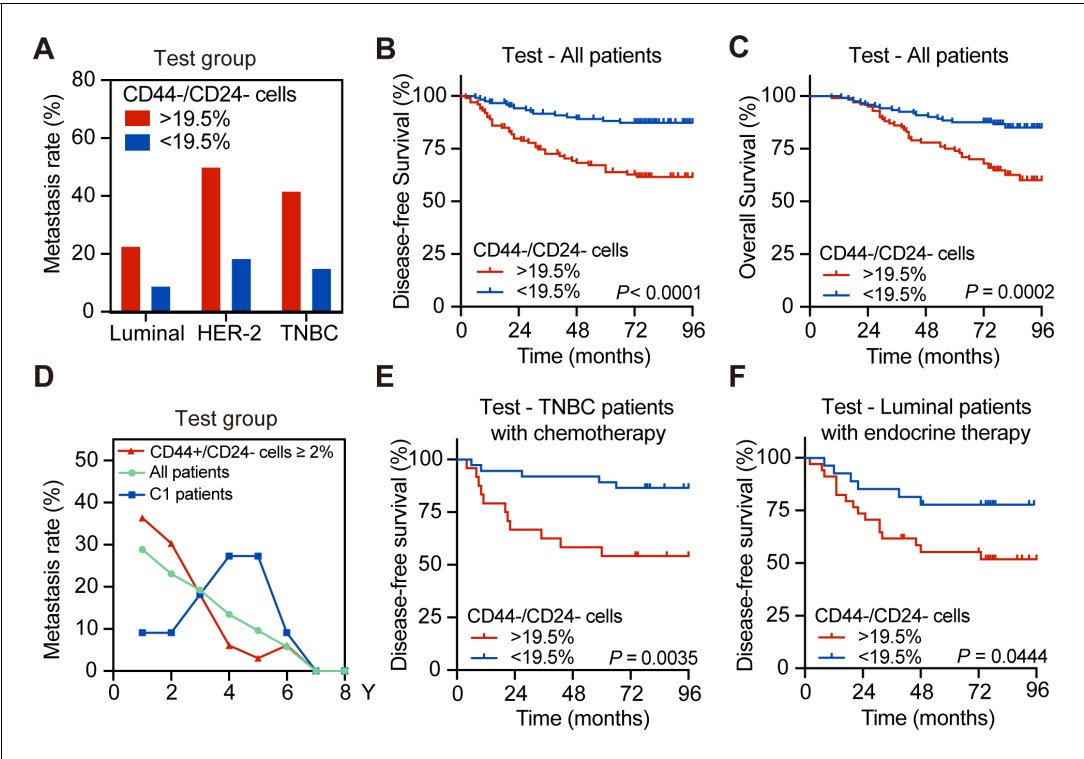

**Figure 2.** The higher frequency of CD44⁻/CD24⁻ tumor cells is associated with delayed distant metastasis and worse DFS in the testing group of patients. (**A**) Postoperative metastasis rate in the test group of patients with different molecular subtypes of breast cancer after they were stratified by 19.5% of CD44⁻/CD24⁻ cells. (**B and C**) Kaplan–Meier analysis of the DFS (**B**) and OS (**C**) of all patients in the testing group after they were stratified into ≥19.5% of CD44⁻/CD24⁻ (n = 100) vs. <19.5% of CD44⁻/CD24⁻ tumor cells (n = 121). (**D**) Longitudinal measurements of metastasis rates among all breast cancer patients (n = 211), patients with ≥2% CD44⁺/CD24⁻ cells (n = 68) and the C1 group of patients with <2% of CD44⁺/CD24⁻ and ≥19.5% CD44⁻/CD24⁻ cells (n = 153). (**E**) Kaplan–Meier analysis of the DFS in the testing group of TNBC patients with chemotherapy after they were stratified into ≥19.5% of CD44⁻/CD24⁻ (n = 24) vs. <19.5% of CD44⁻/CD24⁻ tumor cells (n = 35). (**F**) Kaplan–Meier analysis of the DFS in the testing group of luminal breast cancer patients with endocrine therapy after they were stratified into ≥19.5% of CD44⁻/CD24⁻ (n = 22) vs. <19.5% of CD44⁻/CD24⁻ tumor cells (n = 19). p-Values were determined by log-rank test.

The online version of this article includes the following source data and figure supplement(s) for figure 2:

**Source data 1.** Source data for *Figure 2*.

**Figure supplement 1.** Kaplan–Meier and the log-rank test analyses of DFS and OS in the testing group of breast cancer patients stratified by percentage of CD44⁻/CD24⁻ cells.

**Figure supplement 1—source data 1.** Kaplan–Meier analysis data of *Figure 2—figure supplement 1*.

CD44⁻/CD24⁻ cells could convert into CSCs was tested in vitro and in vivo. First, primary human tumor cells were isolated from TNBC breast cancer patients and primary CD44⁺/CD24⁻ CSCs and CD44⁻/CD24⁻ tumor cells were purified by flow cytometry sorting (*Figure 3A*, *Figure 3—figure supplement 1A*). Subsequently, CD44⁻/CD24⁻ cells were cultured for 7 days. There were 4.6% of CD44⁺/CD24⁻ CSCs (*Figure 3B*). Following implantation with CD44⁻/CD24⁻ MDA-MB-231 cells in the breast fat pad of female BALB/c nude mice, 10.8–16.3% of CD44⁺/CD24⁻ CSCs were detected in the formed tumors at 7 and 21 days post implantation in mice, respectively (*Figure 3C*). Clearly, CD44⁻/CD24⁻ TNBC cells effectively converted into CD44⁺/CD24⁻ CSCs in vitro and in vivo. Because TNBC is the most aggressive type of breast cancer, the spontaneous conversion of CD44⁻/CD24⁻ TNBC cells into CD44⁺/CD24⁻ CSCs was further tested in TNBC cell lines (*Figure 3D* and *Figure 3—figure supplement 1*). CD44⁺/CD24⁻ parent CSCs and CD44⁻/CD24⁻ cells were purified from MDA-MB-231 and MDA-MB-468 cells by flow cytometry sorting and CD44⁻/CD24⁻ cells were cultured in both SC and L15 media for 7 days, respectively. Following culture of CD44⁻/CD24⁻ cells from MDA-MB-231 cells, flow cytometry analysis exhibited 28.3% and 24.7% of CD44⁺/CD24⁻ cells in SC and L15 media, respectively (*Figure 3E*). Similar results were observed for the parental MDA-MB-468 cells (*Figure 3—figure supplement 1B*).

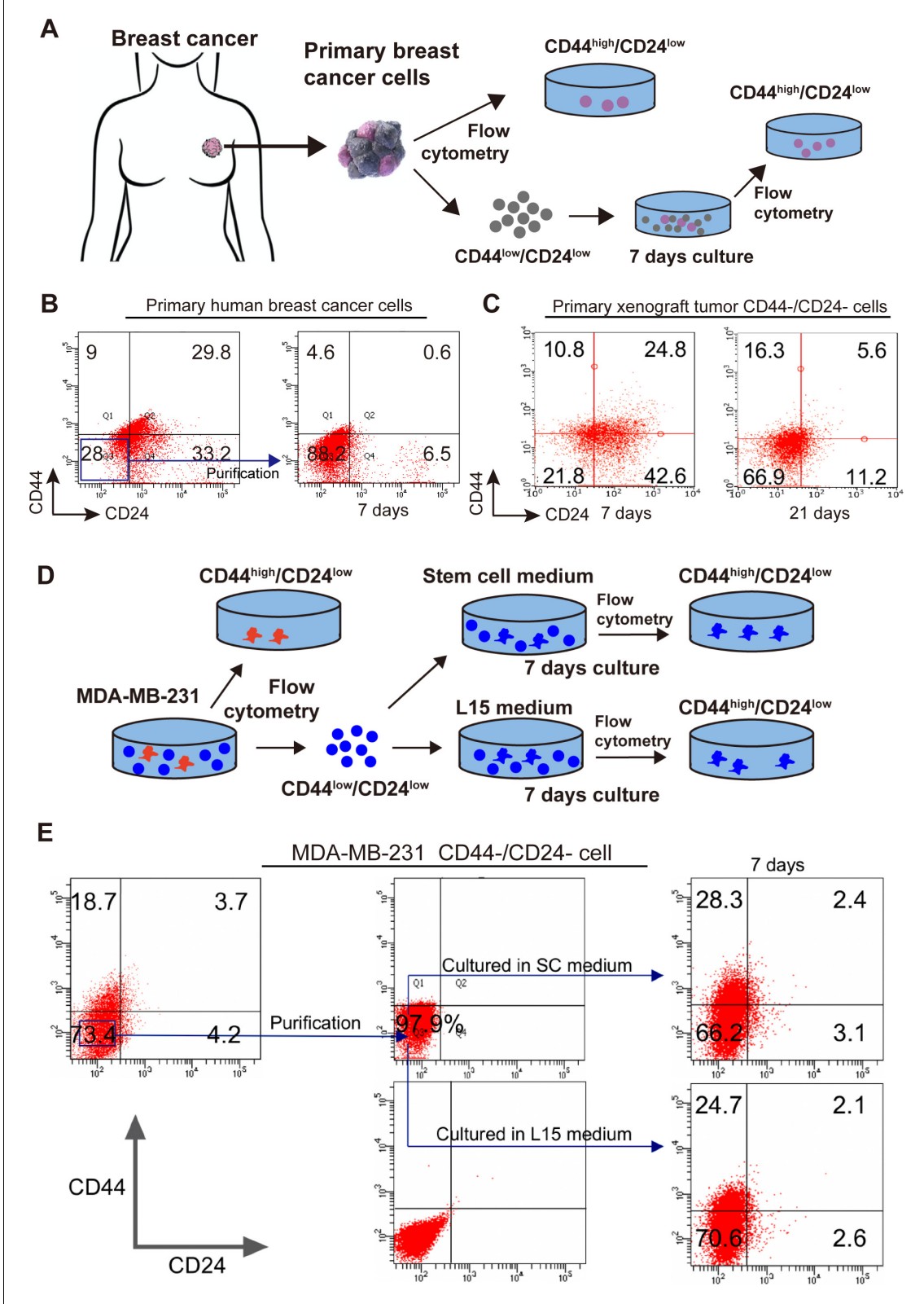

**Figure 3.** Spontaneous conversion of CD44⁻/CD24⁻ TNBC cells into CD44⁺/CD24⁻ CSCs. (**A**) A diagram Illustrated the experimental protocol for testing the spontaneous conversion of primary human breast cancer CD44⁻/CD24⁻ cells into CD44⁺/CD24⁻ CSCs in vitro. (**B, C**) Flow cytometry analysis of the spontaneous conversion of primary human or xenograft breast cancer CD44⁻/CD24⁻ cells into CD44⁺/CD24⁻ CSCs in vitro. The primary human and xenograft breast cancer CD44⁻/CD24⁻ cells were purified from human fresh TNBC tissue cells or MDA-MB-231 xenograft tissue cells by flow cytometry

*Figure 3 continued on next page*

*Figure 3 continued*

sorting and cultured in L15 medium for 7 and 21 (specifically for cells from xenograft tissue cells) days, respectively. The percentages of CD44$^+$/CD24$^-$ CSCs were analyzed by flow cytometry. (D) A diagram illustrated the experimental protocol for testing the spontaneous conversion of CD44$^-$/CD24$^-$ cells from TNBC cells into CD44$^+$/CD24$^-$ CSCs. (E) Flow cytometry analysis of the percentages of CD44$^+$/CD24$^-$ CSCs. CD44$^-$/CD24$^-$ MDA-MB-231 cells were purified by flow cytometry sorting and cultured in the indicated medium for the indicated duration, followed by flow cytometry analysis. The online version of this article includes the following source data and figure supplement(s) for figure 3:

**Source data 1.** Source data for *Figure 3*.

**Figure supplement 1.** The spontaneous conversion of CD44$^-$/CD24$^-$ TNBC cells into CD44$^+$/CD24$^-$ CSCs in vitro.

**Figure supplement 1—source data 1.** The picture of primary human breast cancer cells.

## Parental TNBC CSCs and newly converted CSCs from CD44$^-$/CD24$^-$ TNBC cells display similar biological behaviors in vitro

Next, the biological behaviors of the newly converted CD44$^+$/CD24$^-$ CSCs from CD44$^-$/CD24$^-$ MDA-MB-231 cells (CD44$^-$/CD24$^-$ CSCs) and parental CSCs directly purified from MDA-MB-231 cells (WT CSCs) were measured for their mammosphere formation, self-renewal, tumor cell differentiation, and CSC stemness marker expression in vitro and their tumorigenicity in vivo. Both WT CSCs and CD44$^-$/CD24$^-$ CSCs formed similar numbers of mammospheres with comparable sizes (*Figure 4A*)

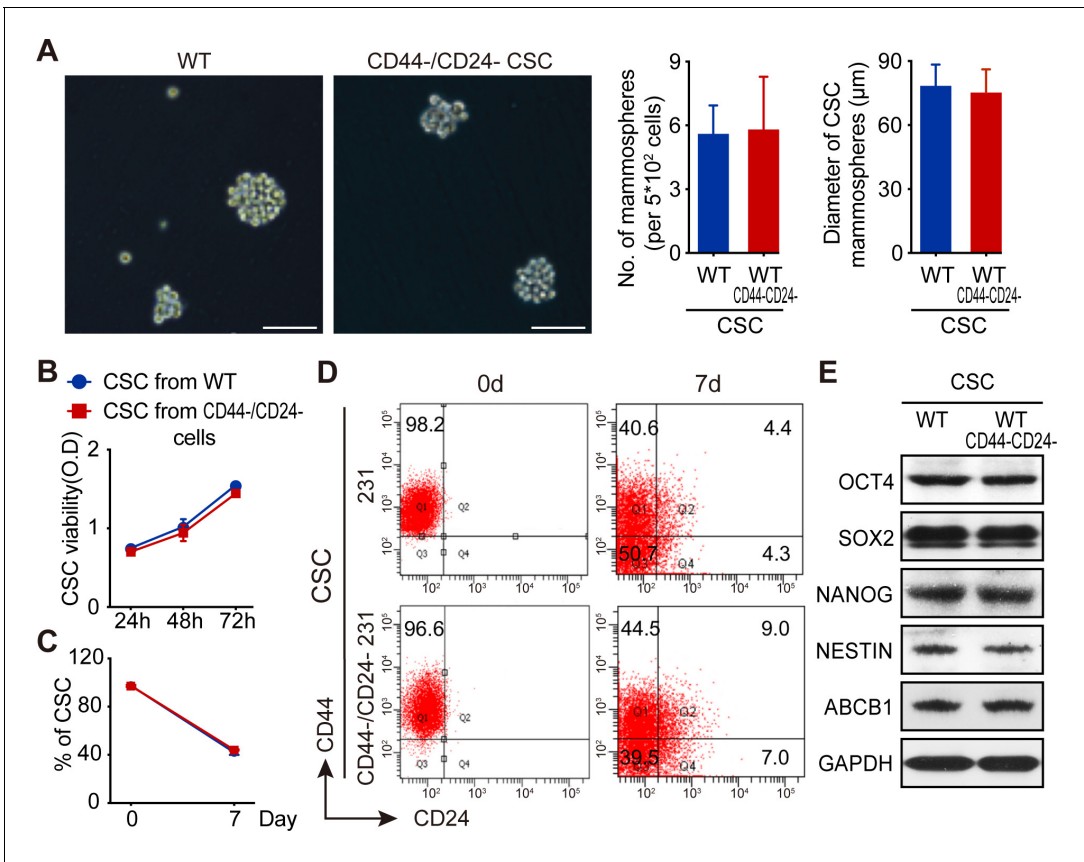

**Figure 4.** Both WT CSCs and CD44$^-$/CD24$^-$ CSCs from TNBC cells have similar biological behaviors in vitro. (A) CD44$^-$/CD24$^-$ CSCs and WT CSCs displayed similar ability to form mammosphere in vitro following culturing them for 7 days. (B) CCK-8 assay analysis of WT and CD44$^-$/CD24$^-$ CSC proliferation. (C) Flow cytometry analysis of the frequency of CSCs after culture WT and CD44$^-$/CD24$^-$ CSCs in SC medium for 7 days. (D) Flow cytometry analysis of WT and CD44$^-$/CD24$^-$ CSC differentiation after culturing them in SC medium for 7 days. Data are representative images or expressed as the mean or mean ± SD of each group from three independent experiments. (E) Western blot analysis of stemness marker expression in WT and CD44$^-$/CD24$^-$ CSCs.

The online version of this article includes the following source data for figure 4:

**Source data 1.** Source data for *Figure 4*.

and displayed similar proliferative capacity (*Figure 4B*). Culture of both types of CSCs for 7 days promoted their differentiation into different subtypes of MDA-MB-231 cells with similar percentages (*Figure 4C,D*). Western blot analysis revealed that the relative levels of OCT4, SOX2, NANOG, NESTIN, and ABCB1 proteins were also comparable between these two types of CSCs (*Figure 4E*). Thus, both types of CSCs exhibited comparable capacities to self-renew, differentiate and form mammospheres, and had similar stemness properties.

### Both WT CSCs and CD44⁻/CD24⁻ CSCs from TNBC cells have similar tumorigenesis and distant metastasis properties in vivo

Next, whether both types of CSCs functioned similarly was tested in *BALB/c nude mice*. After implantation with the same number of each type of CSCs for 21 days, both types of CSCs induced xenograft tumors with similar sizes and weights (*Figure 5A*). Flow cytometric analysis of single cells from these xenograft tumors exhibited similar percentages of different subtypes of breast cancer cells (*Figure 5B*). Furthermore, intravenous injection of equal number of each type of CSCs induced

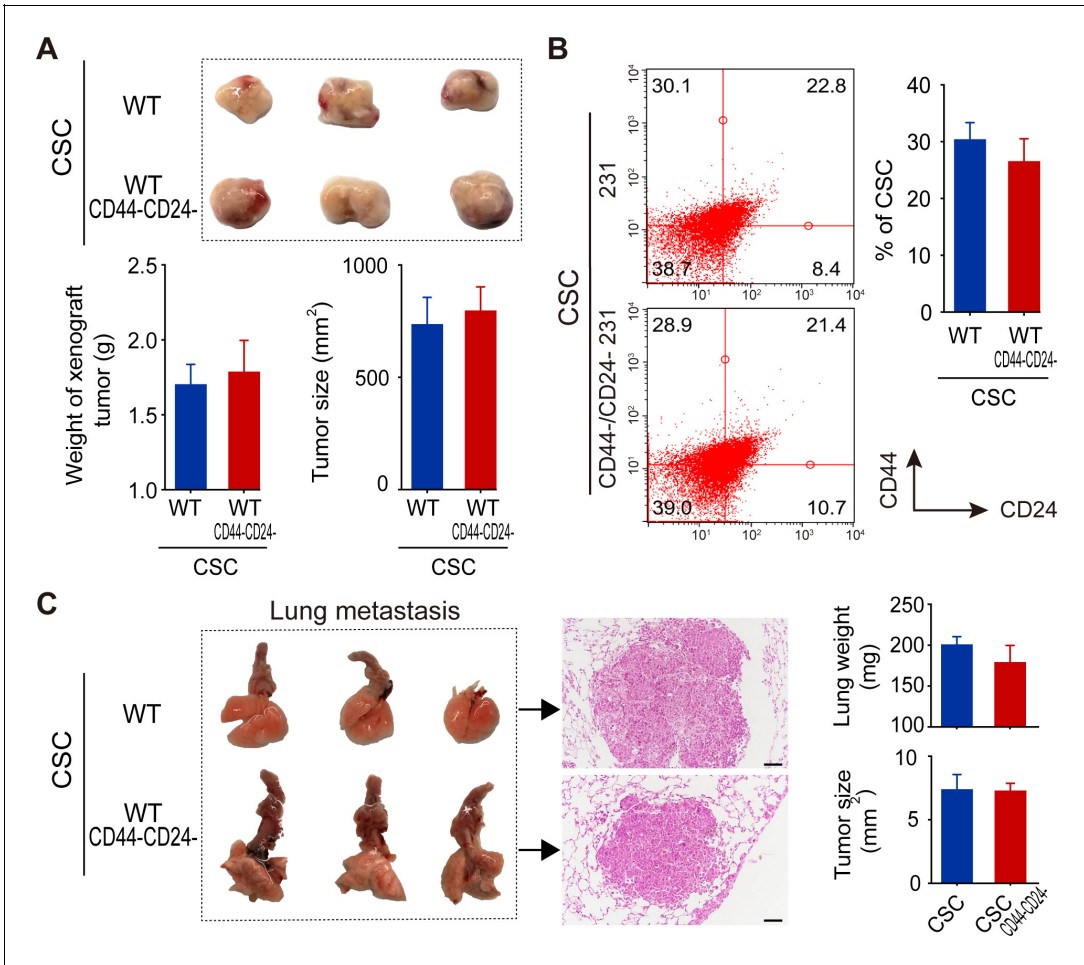

**Figure 5.** Both WT and CD44⁻/CD24⁻ CSCs from MDA-MB-231 cells exhibit similar tumorigenicity and comparable abilities to differentiate and induce lung metastatic in vivo. (**A**) Both WT and CD44⁻/CD24⁻ CSCs had similar tumorigenicity to induce comparable sizes of tumors in BALB/c nude mice following subcutaneous implantation for 21 days (*n* = 5–8 per group). (**B**) Flow cytometry analysis of different subtypes of TNBC cells in xenograft tumors. (**C**) Lung metastasis. NOD/SCID mice were intravenously injected with the same number of WT or CD44⁻/CD24⁻ CSCs and 21 days later, the mice were euthanized and their lungs were dissected for weighing and histological H and E staining to examine lung metastatic morphology and nodule sizes. Data are representative images or expressed as the mean ± SD of each group (*n* = 5–8 per group), and samples were analyzed from three independent experiments.

The online version of this article includes the following source data for figure 5:

**Source data 1.** Source data for *Figure 5*.

lung metastatic nodules with similar tumor sizes and lung weights in *NOD/SCID mice* (*Figure 5C*). H and E staining of lung tissue sections revealed similar pathological characters in both groups of mice (*Figure 5C*). Thus, both WT CSCs and CD44⁻/CD24⁻ CSCs from TNBC cells had similar tumorigenicity and metastatic capacities in vivo.

## *RHBDL2* is crucial for spontaneous conversion of CD44⁻/CD24⁻ breast cancer cells into CD44⁺/CD24⁻ CSCs

To understand the molecular mechanisms underlying spontaneous conversion of TNBC CD44⁻/CD24⁻ cells, CD44⁻/CD24⁻ cells were first purified from MDA-MB-231 cells by flow cytometry and transfected with pGL3-OCT4-EGFP using Lipofectamine 3000. The cells were cultured into micrograft plates in a single cell manner for 1, 3, and 5 days (*Figure 6A*). Subsequently, the cultured individual cells were captured and subjected to RNA-seq analysis to identify DEGs.

There were 11 DEGs associated with tumor cell differentiation and dedifferentiation between EGFP- CD44⁻/CD24⁻ cells and EGFP+ CSCs and they included *RHBDL2, HIST1H4H, DSCC1, ZNF710, ATP8B3,* and others. Particularly, comparison of EGFP+ CSCs (120 hr post culture) with EGFP- (24 hr post culture) and uncommitted cells (72 hr post culture) revealed that both *RHBDL2* and *HIST1H4H* mRNA transcripts significantly increased (*Figure 6B*). GO and KEGG analyses revealed that all DEGs were predominantly involved in cell organelle formation, metabolism, signal transduction, and transcriptional regulation (*Figure 6—figure supplement 1A,B*). Furthermore, Kaplan-Meier analysis and log-rank test indicated that higher levels of *HIST1H4H, RHBDL2, DSCC1, ARL6IP1, PPME1,* and lower levels of *G2E3 and MED22* mRNA transcripts, but not others, were significantly associated with worse DFS in breast cancer patients in the Cancer Genome Atlas (TCGA) database (*Figure 6C*). Finally, the levels of mRNA transcripts of these DEGs among the purified parental CD44⁺/CD24⁻ CSCs (red color), the newly converted CD44⁺/CD24⁻ CSCs (blue color) and the unconverted CD44⁻/CD24⁻ tumor cells (green color) from MDA-MB-231 cells were tested by RT-qPCR. Compared with the unconverted CD44⁻/CD24⁻ tumor cells, *PPME1, RHBDL2* and *HIST1H4H* mRNA transcripts increased in the newly converted CD44⁺/CD24⁻ CSCs with a change of >two *folds*, which were similar to that in parental CD44⁺/CD24⁻ CSCs (*Figure 6D*). Together, these data suggest that up-regulated expression *of these genes* may be crucial for the spontaneous conversion of CD44⁻/CD24⁻ TNBC cells into CD44⁺/CD24⁻ CSCs. Given that *RHBDL2* mRNA transcripts at 120 hr post culture were the highest among the different time points post culture, the following experiments centered on the role of *RHBDL2* in the spontaneous conversion of CD44⁻/CD24⁻ TNBC into CD44⁺/CD24⁻ CSCs and their malignant behaviors.

## *RHBDL2* silencing inhibits the YAP1/UPS31/nuclear factor (NF)-κB signaling and spontaneous CD44⁻/CD24⁻ cell conversion into CD44⁺/CD24⁻ CSCs

Next, how *RHBDL2* and the related signaling affected in the spontaneous CD44⁻/CD24⁻ cell conversion into CD44⁺/CD24⁻ CSCs was explored by silencing *RHBDL2* expression in TNBC cells using siRNA-based technology. Because the YAP1 signaling can suppress USP31 expression, a potent inhibitor of the NF-κB signaling, how *RHBDL2* silencing could affect the relative levels of YAP1 expression and phosphorylation was investigated in MDA-MB-231 and MDA-MB-468 cells. Compared with the controls, *RHBDL2* silencing dramatically decreased RHBDL2 and YAP1 expression and slightly increased YAP1 phosphorylation in both TNBC cells (*Figure 7A*). Furthermore, *RHBDL2* silencing also obviously increased the relative levels of USP31 expression, YAP1 phosphorylation and decreased the relative levels of NF-kB phosphorylation, besides the demolished RHBDL2 expression, in CD44⁻/CD24⁻ cells sorted from both the negative control (NC) and *RHBDL2*-silenced MDA-MB-231 cells and MDA-MB-468 cells (*Figure 7B*). Moreover, *RHBDL2* silencing not only decreased YAP1 expression in the cytoplasm, but also dramatically reduced the levels of nuclear YAP1 in both TNBC cells (*Figure 7C*). In addition, after culture of control CD44⁻/CD24⁻ and *RHBDL2*-silenced CD44⁻/CD24⁻ cells in L15 medium for 7 days, the results indicated that *RHBDL2* silencing remarkably decreased the percentages of CD44⁺/CD24⁻ CSCs from 6.5–5.5% to 0.5–0.7% in both types of TNBC cells (*Figure 7D,E*). Finally, we determined if *RHBDL2* silencing could modulate the mammosphere formation of CD44⁺/CD24⁻ CSCs in vitro. CD44⁻/CD24⁻ cells were purified from MDA-MB-231 and MDA-MB-468 cells and transfected with control siRNA or *RHBDL2*-specific siRNA, followed

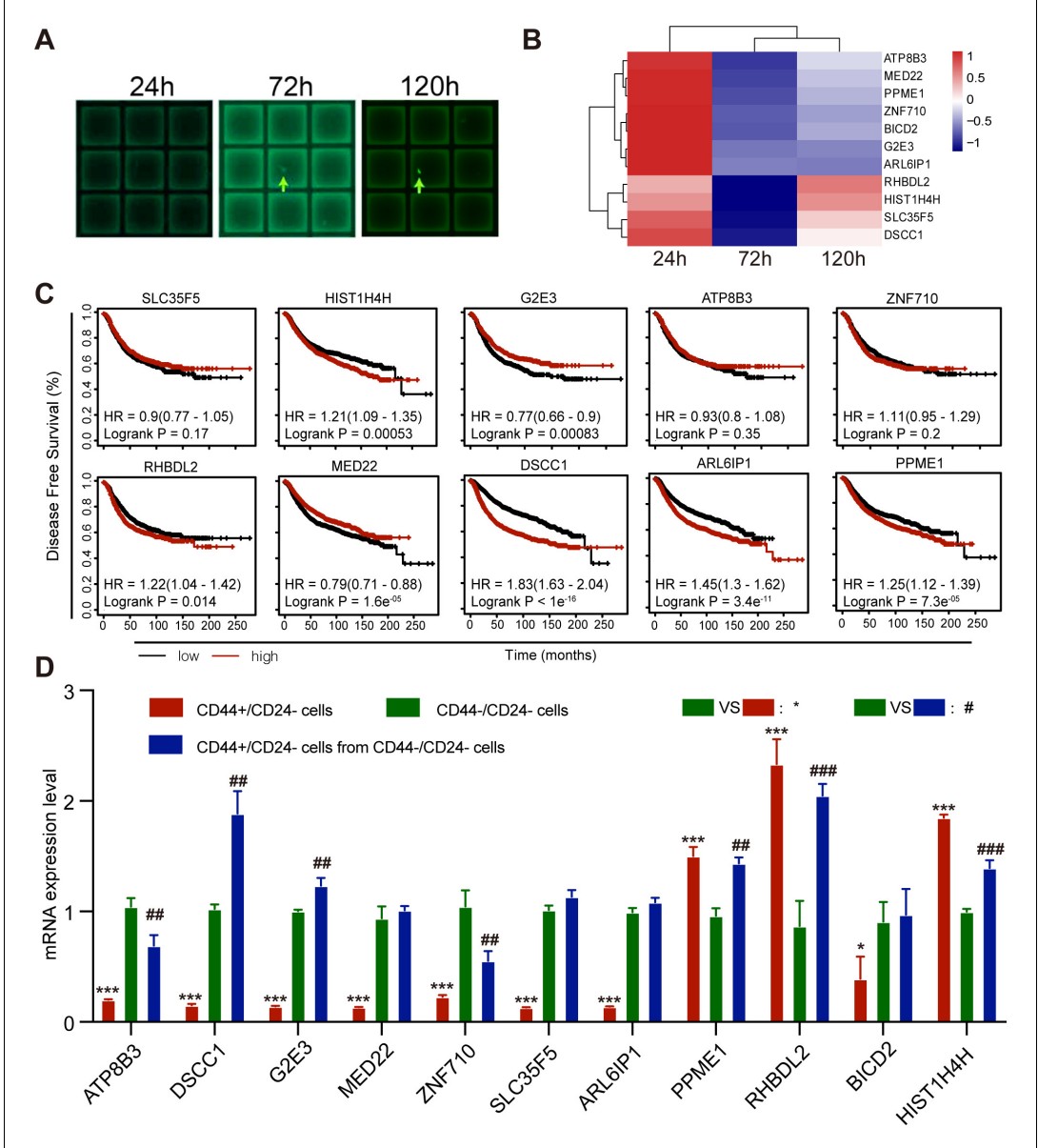

**Figure 6.** *RHBDL2* expression is up-regulated during the process of CD44−/CD24− MDA-MB-231 cell conversion into CD44+/CD24− CSCs. (**A**) Representative images of micrograft plates for culture of CD44−/CD24− MDA-MB-231 cells at a single cell level. CD44−/CD24− MDA-MB-231 cells were purified by flow cytometry sorting and transfected with pGL3-OCT4-EGFP, followed by culturing them in micrograft plates at a single cell level for 24, 72, and 120 hr, and the average of the gene expression profile of three cells are photographed. (**B**) RNA-seq analysis and heatmap displayed the top DEGs during the process of CD44−/CD24− MDA-MB-231 cell conversion into CD44+/CD24− CSCs. (**C**) Kaplan-Meier estimation of the association of the expression of DEGs with DFS in breast cancer patients in TCGA database. (**D**) RT-qPCR analysis of the relative levels of gene mRNA transcripts in WT CD44+/CD24−, CD44−/CD24− CSCs, and unconverted CD44−/CD24− cells after culture for 7 days. Data are representative images or expressed as the mean ± SD of each group from at least three separate experiments. *p<0.05, ***p<0.001; ##p<0.01, ###*Pp*<0.001 vs. the CD44−/CD24− cells, determined by Student's *t*-test.

The online version of this article includes the following source data and figure supplement(s) for figure 6:

**Source data 1.** Source data for *Figure 6*.

**Figure supplement 1.** Bioinformatic analysis of DEGs between 24 hr converted CSCs and CD44−/CD24− MDA-MB-231 cells.

**Figure supplement 1—source data 1.** The data of bioinformatic analysis.

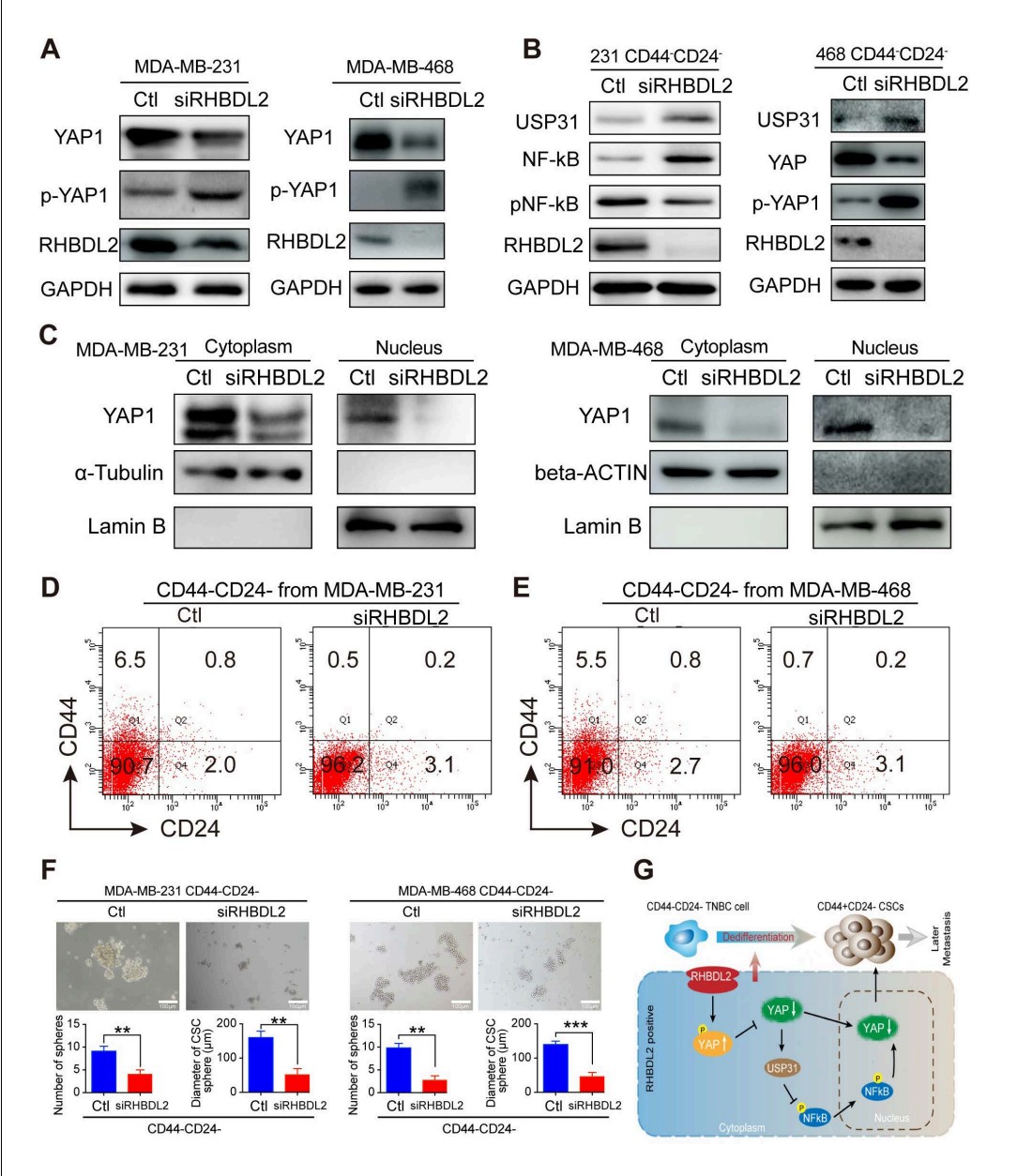

**Figure 7.** *RHBDL2* silencing inhibits the spontaneous conversion of CD44⁻/CD24⁻ cells into CSCs by attenuating the YAP1/NF-kB signaling through enhancing USP31 expression in TNBC cells. (**A**) Western blot analysis exhibited that *RHBDL2* silencing decreased RHBDL2 and YAP1 expression, but increased YAP1 phosphorylation in TNBC cells. (**B**) Western blot analysis displayed that *RHBDL2* silencing decreased RHBDL2 and YAP1 expression, NF-kB activation, but increased USP31 expression and YAP1 phosphorylation in the indicated TNBC CD44⁻/CD24⁻ cells. (**C**) Western blot analysis of cytoplasmic and nuclear YAP1 protein levels in MDA-MB-231/MDA-MB-468 cells following *RHBDL2* silencing indicated that *RHBDL2* silencing mitigated the nuclear translocation of YAP1 in TNBC cells. (**D and E**) *RHBDL2* silencing inhibited the spontaneous conversion of CD44⁻/CD24⁻ cells into CSCs in vitro. CD44⁻/CD24⁻ cells were purified from MDA-MB-231 (**D and E**) *RHBDL2* silencing inhibited the spontaneous conversion of CD44⁻/CD24⁻ cells into CSCs in vitro. CD44⁻/CD24⁻ cells were purified from MDA-MB-231 (**D**) and MDA-MB-468 (**E** ) cells and transfected with the control or *RHBDL2*-specific siRNA, followed by culturing them in L15 medium for 7 days. Subsequently, the percentages of CD44⁺/CD24⁻ CSCs in each group of cells were analyzed by flow cytometry. (**F**) *RHBDL2* silencing attenuated mammosphere formation of CD44⁻/CD24⁻ CSCs. Following transfected with the control or *RHBDL2*-specific siRNA, CD44⁻/CD24⁻ CSCs were cultured in L15 medium for 7 days and the formed mammospheres were photoimaged (magnification x 400) and their numbers and sizes were measured in a blinded manner. Data are representative images or expressed as the mean ± SD of each group from at least three separate experiments. *p <0.05, **p <0.01, determined by Student *t*-test. (**G**) A diagram illustrates the possible mechanisms underlying the action of RHBDL2 in enhancing the spontaneous conversion of CD44⁻/CD24⁻ cells into CD44⁺/CD24⁻ CSCs, contributing to delayed distant metastasis of breast cancer.

*Figure 7 continued on next page*

*Figure 7 continued*

The online version of this article includes the following source data for figure 7:

**Source data 1.** Source data for *Figure 7*.

by culturing them in L15 medium for 7 days. Subsequently, their ability to form mammospheres was examined. Compared with the control siRNA-transfected cells, *RHBDL2* silencing significantly decreased the numbers and sizes of formed mammospheres in both types of TNBC cells (*Figure 7F*). Therefore, these findings further suggest that RHBDL2 may be a crucial regulator of the spontaneous conversion of CD44⁻/CD24⁻ TNBC cells into CD44⁺/CD24⁻ CSCs by enhancing the YAP1/USP31/NF-κB signaling in breast cancer cells.

## Discussion

The current study uncovered that CD44⁻/CD24⁻ TNBC cells were able to spontaneously convert to CD44⁺/CD24⁻ CSCs, consistent with previous studies in vitro (*Italiano and Shivdasani, 2003*; *Zoppino et al., 2010*) both WT CSCs and CD44⁻/CD24⁻ CSCs had similar biological properties in terms of stemness marker expression, self-renewal, differentiation, tumorigenicity and lung metastasis in vitro and in vivo a higher frequency (≥19.5%) of CD44⁻/CD24⁻ cells in breast cancer tissues was associated with worse DFS and delayed postoperative distant metastasis a higher frequency of CD44⁻/CD24⁻ and CD44⁺/CD24⁻ cells, higher tumor N stage, and molecular subtypes were predictors of DFS for breast cancer patients; and mechanistically, *RHBDL2* silencing inhibited the YAP/USP31/NF-κB signaling and attenuated the spontaneous CD44⁻/CD24⁻ cell conversion into CD44⁺/CD24⁻ CSCs and their mammosphere formation. Therefore, the findings may uncover new biomarkers for prognosis of delayed distant breast cancer metastasis and shed light on the molecular mechanisms underlying the distant metastasis of breast cancer, particularly for TNBC. It is necessary to further investigate whether RHBDL2 can be a therapeutic target in inhibiting the development and progression of breast cancer.

Previous studies have shown that non-CSC tumor cells can dedifferentiate (convert) into CSCs in human glioblastoma and intestinal stroma melanoma, contributing to intra- and inter-tumor heterogeneity (*Stepanova et al., 2003*; *Tzimas et al., 2006*; *Wei et al., 2019*). Furthermore, CSC plasticity and heterogeneity can promote tumor progression and resistance to therapy (*Das et al., 2020*; *Kilmister et al., 2020*; *Martin-Castillo et al., 2013*; *Thankamony et al., 2020*). For example, intra-tumoral heterogeneity is a major ongoing challenge for effective cancer therapy, while CSCs are responsible for intra-tumoral heterogeneity, therapeutic resistance, and metastasis, which may be because cancer cells exhibit a high level of plasticity and an ability to dynamically switch between CSC and non-CSC states or among different subsets of CSCs (*Thankamony et al., 2020*). Consistently, the differentiated non-CSCs can revert to be trastuzumab-refractory, CSS-like cells by enhancing their epithelial-to-mesenchymal transition process (*Martin-Castillo et al., 2013*). Similarly, the changes in tumor microenvironment and epigenetics can promote the conversion of non-CSCs into CSCs, leading to tumor progression and therapeutic resistance (*Das et al., 2020*). The current study further confirmed that CD44⁻/CD24⁻ TNBC cells spontaneously converted into CD44⁺/CD24⁻ CSCs that possessed biological properties, similar to their CSCs isolated directly from parental TNBC cells. More importantly, a higher frequency of CD44⁻/CD24⁻ cells in breast cancer tissues was associated with significantly with worse DFS and delayed distant metastasis. These, together with the dynamic process of spontaneous conversion of non-CSC CD44⁻/CD24⁻ cells and CSC differentiation, indicate that the spontaneous conversion of CD44⁻/CD24⁻ cells helps maintain the CSC pool size, contributing to the delayed distant metastasis and worse prognosis in breast cancer patients.

The data from the current study indicated that a higher frequency of CD44⁻/CD24⁻ cancer cells, like higher frequency of CSCs, was one of the independent risk factors for delayed distant metastasis and worse DFS in this population. Particularly, the higher frequency of CD44⁻/CD24⁻ cancer cells was a better predictor of delayed breast cancer metastasis (up to 12 years after initial breast cancer diagnosis). Evidently, while high frequency of CSCs was valuable for predicting distant metastasis at 5–7 years post-surgery a higher frequency of CD44⁻/CD24⁻ cells effectively predicted delayed distant metastasis in cases with a low frequency of CSCs. These results suggest that CD44⁻/CD24⁻ cells may

spontaneously convert into CSCs and cause delayed distant metastasis when environmental and other factors cause a clonal evolution by accumulating successive mutations (*Meacham and Morrison, 2013*).

In addition, the RNA-seq analysis and subsequent RT-qPCR identified several DEGs during the dynamic process of CD44⁻/CD24⁻ TNBC cell conversion into CSCs, such as *RHBDL2*, and its expression was up-regulated in the newly converted CSCs. The RHBDL2, also known as the rhomboid like 2, is one intramembrane serine protease of the secretase-A rhomboid family (*Etheridge et al., 2013*). Previous studies have identified that the RHBDL2 cleaves its substrates, including EGF, ephrin-β3, thrombomodulin, the C-type lectin family 14 member A (CLEC14A), cadherin, IL-6R, Spint-1 and the collagen receptor tyrosine kinase DDR1 (*Adrain et al., 2011*; *Battistini et al., 2019*; *Etheridge et al., 2013*). Although the functional consequence of cleaving these substrates has not been fully explored the available findings indicate that RHBDL2 functionally activates the EGF signaling, cadherin shedding, angiogenesis and promotes the wound healing and migration of different types of cells, including tumor cells (*Adrain et al., 2011*; *Cheng et al., 2011*). Furthermore, up-regulated RHBDL2 expression is associated with worse prognosis of several types of malignant tumors, such as breast cancer, pancreatic adenocarcinoma, clear cell kidney cancer, and low-grade glioma patient (*Canzoneri et al., 2014*; *Johnson et al., 2017*). However, little is known on how upregulated RHBDL2 expression promotes malignant behaviors of these types of tumors. Interestingly, the current study not only observed that the upregulated *RHBDL2* transcription was associated with worse DFS of breast cancer patients in TCGA, but also found that *RHBDL2* silencing inhibited the YAP1/NF-κB signaling by upregulating USP31 expression and attenuated the spontaneous conversion of CD44⁻/CD24⁻ cells into CD44⁺/CD24⁻ CSCs and their mammosphere formation. Moreover, these data extended previous observation that YAP1-mediated suppression of USP31 enhances NF-κB activity to promote sarcomagenesis (*Kemeny and Fisher, 2018*; *Mehta et al., 2018*). YAP1 can enhance CSC stemness in several types of human cancers, and aberrant YAP1 activation is associated with a low level of TNBC differentiation and poor survival of breast cancer patients (*Bora-Singhal et al., 2015*; *Hansen et al., 2015*). Therefore, the findings from the current study indicated that RHBDL2 promoted the spontaneous conversion of CD44⁻/CD24⁻ cells into CD44⁺/CD24⁻ CSCs by enhancing YAP1 stability through inhibiting its phosphorylation to suppress USP31 expression, enhancing the NF-κB signaling in breast cancer cells, whereas *RHBDL2* silencing had the opposite effects in breast cancer cells (*Figure 7G*). Alternatively, RHBDL2 may cleave other substrates to activate the EGF receptor-mediated signaling, and modify cadherin and other molecules through their involved signal pathways to promote the spontaneous conversion of CD44⁻/CD24⁻ cells into CD44⁺/CD24⁻ CSCs. Hence, these findings may provide new insights into the molecular mechanisms underlying the action of RHBDL2 in regulating the CSC pool to promote the metastasis of TNBC. Given that CSCs are crucial for breast cancer progression and metastasis, RHBDL2 may be a new therapeutic target for control of CD44⁻/CD24⁻ cell conversion into CSCs to reduce CSC pool size and limit breast cancer progression and metastasis.

Definitely, this study had limitations. First, the sample size in some groups remained relatively smaller, which might affect statistical power. Second, this study only centered on the role of RHBDL2, but not other DEGs in regulating the spontaneous conversion of CD44⁻/CD24⁻ TNBC cells into CD44⁺/CD24⁻ CSCs. The results might miss some important information on the molecular mechanisms underlying the spontaneous conversion. Furthermore, the method for identification of CD44⁻/CD24⁻ TNBC cells did not include a positive tumor molecular marker, which might contaminate some other types of cells, although using histological tissue sections for identification of tumor cells. Thus, further studies are warranted with advanced cutting-edge technologies, such as combination of single-cell RNA-seq, multiple marker-based CyTOF mass cytometry or spatial proteomics to quantify the frequency or number of CD44⁻/CD24⁻ breast cancer cells in fresh breast cancer tissues, together with prospectively following-up the patients, in a bigger population to investigate the prognostic value of CD44⁻/CD24⁻ breast cancer cells and the molecular mechanisms underlying the spontaneous conversion of CD44⁻/CD24⁻ TNBC into CD44⁺/CD24⁻ CSCs in the metastasis of breast cancer.

In summary, postoperative breast cancer metastasis, especially delayed metastasis (e.g. ≥5 years after diagnosis), is an important unresolved issue. The results from the current study indicated that patients with a higher frequency of CD44⁻/CD24⁻ cells in their breast cancer tissues had a high risk to develop delayed distant metastasis 5–7 years after diagnosis. Mechanistically, CD44⁻/CD24⁻ cells

spontaneously converted into CD44$^+$/CD24$^-$ CSCs and both WT and newly converted CSCs had similar stemness properties in vitro and in vivo. *RHBDL2* silencing significantly decreased YAP1 expression and increased USP31 expression to attenuate the NF-kB activation and CD44$^-$/CD24$^-$ cell conversion into CSCs as well as their mammosphere formation. RHBDL2 may act as a positive regulator of the spontaneous conversion of CD44$^-$/CD24$^-$ cells into CSCs by enhancing the YAP1/USP31/NF-kB signaling. Alternatively, RHBDL2 may cleave other substrates to activate the EGF and other signal pathways, enhancing the spontaneous conversion of CD44$^-$/CD24$^-$ TNBC into CSCs and their metastasis. These novel findings may provide insights into the molecular process of non-CSCs conversion into CSCs, leading to breast cancer progression and delayed distant metastasis. Therefore, the current findings may uncover a novel biomarker for prognosis and therapeutic target for intervention of breast cancer, particularly for TNBC.

## Acknowledgements

This work was supported by grants from The National Natural Science Foundation of China (#81572609), the Major Project Construction Foundation of China Medical University (#2017ZDZX05) and Liaoning Province young top talent project (#XLYC1807099).

## Additional information

### Competing interests

Caigang Liu: Reviewing Editor, eLife. The other authors declare that no competing interests exist.

### Funding

| Funder | Grant reference number | Author |
| --- | --- | --- |
| National Science Foundation | #81572609 | Caigang Liu |

Funders have no role in research design, data collection and decisions to interpret or submit works for publication.

### Author contributions

Xinbo Qiao, Data curation, Formal analysis, Methodology, Writing - original draft, Writing - review and editing; Yixiao Zhang, Validation, Methodology; Lisha Sun, Resources, Software, Writing - review and editing; Qingtian Ma, Formal analysis, Writing - original draft, Project administration; Jie Yang, Software, Investigation; Liping Ai, Data curation, Formal analysis; Jinqi Xue, Guanglei Chen, Xi Gu, Formal analysis; Hao Zhang, Data curation, Software; Ce Ji, Formal analysis, Funding acquisition; Haixin Lei, Yongliang Yang, Writing - review and editing; Caigang Liu, Conceptualization, Writing - review and editing

### Author ORCIDs

Xinbo Qiao ![ORCID] https://orcid.org/0000-0002-6759-921X
Lisha Sun ![ORCID] https://orcid.org/0000-0002-4095-5026
Caigang Liu ![ORCID] https://orcid.org/0000-0003-2083-235X

### Ethics

Animal experimentation: The experimental protocol was approved by the Animal Research and Care Committee of China Medical University (Shenyang, China) and followed the Guidelines of the Care and Use of Laboratory Animals issued by the Chinese Council on Animal Research. Female BALB/c nude mice (6 weeks old) were obtained from Human Silaikejingda Laboratory Animals (Changsha, China) and housed in a specific pathogen-free facility with free access to autoclaved food and water. All surgery was performed under sodium pentobarbital anesthesia, and every effort was made to minimize suffering. The current study was approved by the Ethics Committee of all three hospital review board review boards ((Project identification code: Project identification code: 2018PS304K, date on 03/05/2018 2018PS304K, date on 03/05/2018)).

Decision letter and Author response
Decision letter https://doi.org/10.7554/eLife.65418.sa1
Author response https://doi.org/10.7554/eLife.65418.sa2

## Additional files

### Supplementary files

• Source data 1. Clinical information for 576 patients.

• Supplementary file 1. Clinicopathological characteristics of patients.

• Supplementary file 2. The table of Youden Index.

• Supplementary file 3. Univariate and multivariate Cox regression analyses of clinicopathological factors as predictors of DFS.

• Supplementary file 4. Primary antibodies used in western blotting.

• Transparent reporting form

### Data availability

All data generated or analyzed during this study is included in the manuscript and supporting documents.

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
