## [Decision Letter]

**Acceptance summary:**

The manuscript describes a possible link between the presence of CD44-/CD24- cells and post-operative site distant metastasis. The functional significance of CSC, particularly of the phenotype CD44-/CD24- remains elusive and is worthy of careful evaluation. The strength of the study is the inclusion of a large number of patient samples, which were used in a retrospective analysis. The resulting data provide the rationale for the application of contemporary technologies in the evaluation of CDX and PDX models. As such, this work paves the way for further investigation at a mechanistic and translational level, which may lead to a better understanding of breast cancer metastasis.

**Decision letter after peer review:**

Thank you for submitting your article "Association of CD44 -/CD24 -Breast Cancer Cells with Late Stage Tumor Recurrence" for consideration by *eLife*. Your article has been reviewed by 2 peer reviewers, and the evaluation has been overseen by a Reviewing Editor and Mone Zaidi as the Senior Editor. The following individual involved in review of your submission has agreed to reveal their identity: Robin Anderson (Reviewer #1).

The reviewers have discussed their reviews with one another, and the Reviewing Editor has drafted this to help you prepare a revised submission. Essential revisions:

1) Referees comments must be addressed in a convincing manner, with additional data.

2) Edits are required in all sections of the manuscript.

3) If extensively revised, this manuscript would represent an incremental contribution in the field of breast cancer biology.

*Reviewer #1:*

The authors set out to demonstrate the importance of a subset of cancer cells lacking CD44 and CD24, in promoting tumor progression, metastasis and resistance to therapy. They analyse archival breast tissues from patients where the clinical outcome – development of metastasis and therapy response – are known. They then take this information into cell culture experiments, seeking the mechanism that underlies the importance of these CD44-/CD24- cells. Their major premise is that these double negative cells can de-differentiate into cancer stem cells that subsequently promote metastasis and chemotherapy resistance.

The overall concept of this study is good and the cell-based results are mainly clear. However, the archival tissue analysis needs clarification, as does the RNAseq analysis, as detailed below.

The breast cancer tissues are scored by immunofluorescence for expression of CD44 and CD24, but there is no indication that these two markers are being scored only in tumor cells rather than in all lineages within the tumor microenvironment. This is an important point that must be clarified. Also, the RNAseq analysis is reported to be conducted on single cells, but the data presented do not indicate if this is actually the case and how many replicate samples were analysed and which sample yielded each heat map pattern. Again clarification is required before the data can be interpreted.

Finally, the conclusion from this study is that RHBDL2 induces spontaneous conversion of CD44-/CD24- cells into CD44+/CD24- cells via YAP1, USP31 and decreased NFkB signaling. This may well be the case, but has not been proven by the experiments presented here. Whilst DHBDL2 does alter YAP, USP31 and NFkB, it could also be working through an alternate pathway to induce CSCs.

1. Figure 1 and S1 and S2A score the numbers of samples with the varying CD44/CD24 protein levels. For Figure S1, this is shown by immunofluorescence, but there is no indication of whether the cells being analysed are tumor cells or host cells from the microenvironment. There should be co-staining for EpCAM or CK19 to confirm that tumor cells only are being scored. I am assuming that the data shown in Figure S2A are actually from the immunostaining and not from flow cytometry analysis as stated in the figure legend? If this was a flow analysis, there is no description of the origin or processing of the samples reported in Figure 1A for flow cytometry. The legend for Figure S2A states that the data are stratifying only metastatic tumors, but in Figure 1A, apparently many of the tumors showed no metastasis. All these points need to be clarified/corrected.

2. This first section of the manuscript is poorly presented and I am struggling to follow the arguments and data being provided. Many of the figures are mis-labelled in the text and they are not presented in consecutive order as discussed in the text. For some references to data in the text where the wrong figure has been cited, I cannot even find the correct figure.

3. Possibly the way the data are presented, with some figures referring to the training set and other figures to the test set is adding to the confusion when it is not clearly stated which set is under discussion at the time.

4. Late metastasis – better phrased as delayed metastasis.

5. The sentence in the Introduction: "Previous studies have shown that CD44+/CD24- breast CSCs might be a dominant factor in relapse of triple negative breast cancer (TNBC), due to their possession of potent self-renewal and differentiation capacities to differentiate into mature CD44-/CD24-, CD44+/CD24+, and CD44-/CD24+ cancer cells (Geng et al., 2014; Wang et al., 2014)". It is not at all clear to me that the references provided support this comment. Geng et al. do not mention CD44-/CD24- cells at all, and Wang et al. do not discuss differentiation of CD44+/CD24- into the other phenotypes. Only CD44- cells are mentioned, not CD24.

6. Sentence in Introduction: "Indeed, injection of up to 1000 breast CSCs was able to generate a solid tumor mass in immunocompromised mice (Chaffer et al., 2011; Iliopoulos et al., 2011)." Please clarify. It is not obvious to me that Chaffer et al. specified how many CSCs they injected and Iliopoulos et al. showed that as few as 50 CSCs could form tumors.

7. Sentence in Introduction: "Moreover, previous studies reported that non-CSCs, such as CD44-/CD24- TNBC cells, are able to spontaneously convert into CSCs to renew the CSC pool, resulting in chemoresistance (Gruber et al., 2016; Kim et al., 2015; Ye et al., 2018)." The statement being referenced is for non-CSCs in TNBC, but Gruber et al. and Ye et al. are not reporting on TNBC.

8. Figure 1B – please explain more fully how the cut-off of 19.5% CD44-/CD24- was determined.

9. Figure 1C – please explain how the ratios of different subtypes of breast cancer with high or low CD44-/CD24- were calculated. According to the figure, for luminal cancers, there was a metastasis rate of ~40% in the high group and less then 10% in the low group. How were the numbers of 63.1% and 32.6% derived from Figure 1C?

Then the same figures are provided for Figure 2A, when using the test set. Were the data in Figure 1C from the training set? Why are the final metastasis rates exactly the same? This needs a better explanation.

10. Page 6: reference to Figure S3C should be Figure S2C?

11. Page 7 – Should Figure S5A,B actually be Figure S3?

12. Page 7: The statement: "Moreover, a low CSC percentage led to different risks for developing tumor metastasis 5 years after diagnosis and adjuvant chemotherapy between the C1 and C0 patients (Figure S3D)." This does not appear to apply to Figure S3D and I cannot tell to which figure this statement is referring.

13. Page 7, last sentence: "In the present study, we first designed the experiments illustrated in Figure 3A to characterize the percentages of different cell subtypes among MDA-MB-231 cells using cell culture and flow cytometric cell sorting of parental MDA-MB-468 cells." If I understand correctly, this should re-worded as follows: "In the present study, we first designed the experiments illustrated in Figure 3A and Figure S5 to characterize the percentages of different cell subtypes among MDA-MB-231 and MDA-MB-468 cells using cell culture and flow cytometric cell sorting of parental cells.

14. Figure 3C raises questions about reproducibility of the data. The profile in Figure 3C after 7 days in culture looks very different from that shown in Figure 3B.

15. Figure 5C: please explain how you obtained a weight measurement for the lung metastases. You record lung metastasis weights of up to 2 grams. Lungs typically weight about 0.2 grams. Lungs of 2 grams seem improbable, let alone metastases in lungs weighing 2 grams. Also, tumor sizes of 700 mm2 do not seem possible in lungs.

16. The RNAseq analysis requires more explanation. In Figure 6B, please explain the three columns shown in the heat map. Without their designation, the rest of this figure is hard to interpret. It seems that only one single cell per timepoint has been analysed. Is this correct? How reliable is sequencing from a single cell? Or was the analysis from many cells grown up from a single cell? How many single cells (or derivative of single cells) were analysed at each time point? There is a discrepancy between the protocol described in the text and that provided in the Methods section, when describing the RNAseq analysis. It is difficult to relate the significance of the genes shown in Panel B to the Kaplan-Meier data when it is not clear what the three columns in panel B represent.

17. Figure 6D: I assume that the "*" and "#" are meant to convey some sort of significance, but they do not appear for any of the genes assessed by RT-PCR. Does that mean that none of the changes were significant?

18. Why does the legend for Figure S6 talk about SILAC for protein analysis? Was SILAC run on these samples? Impossible on single cells. No mention of SILAC in Methods.

19. Page 10: Explain why YAP was selected for analysis following knockdown of RHBDL2. Was there a previously known connection between these two genes? For the CD44-/CD24- cells in Figure 7B, was YAP phosphorylation also altered? Please show a western demonstrating that RHBDL2 is reduced after transfection with siRHBDL2.

20. Figure 7D: text says 0.7% CD44+/CD24- but figure says 0.5% and vice versa for the MDA-MB-468 cells.

21. The text on page 11 needs re-writing. The second paragraph, commencing with "We next selected…" is a repeat of the text above where you have already done this step and analysed the cells by flow cytometry.

*Reviewer #2:*

The manuscript describes a possible link between the presence of CD44-/CD24- cells and post-operative site distant metastasis. This is not a new concept, yet the actual biological function of CSC, particularly of the phenotype CD44-/CD24- remains inconclusive. The primary strength of the study is the large number of patient samples used in the retrospective analysis that provides the rationale for in vitro experiments using cultured cancer cells. Several technical and methodological limitations were identified.

The description of the patient cohort is very elusive. The bioinformatic analysis lacks in rigor as the methodology used to identify and quantitate CD44-/CD24- cells in patients is not properly described. For instance, Figure legends describe the use of flow cytometry for quantification of CD44-/CD24- negative cells from patient samples. Yet only an immunofluorescence staining is described in Results. No Methods description is given to FACS of patient samples. The manuscript seems focused on TNBC, yet the reader has a difficult time understanding how many samples are actually from TNBC patients. Two TNBC established cell lines are mentioned in the manuscript but no rationale is provided as to why some experiments are performed only with MDA-MB-468 and some with MDA-MB-231. The same is true for several other experiments. The extensive molecular description of CD44-/CD24- cell dedifferentiation into CSCs deserves distinction but will need to be confirmed in patient samples before conclusions can be drawn with regards to the biological significance of these findings to breast cancer patients and the medical community at large.

Clarify the methodology used to quantitate CD44-/CD24- in patient samples, as well as the bioinformatic analysis supporting the current hypothesis.

Provide a detailed analysis of clinical and pathologic information of the patient cohort, including histology.

Several references are 10 years old or older, new references supporting the authors finds would strength the hypothesis.

---

## [Author Response]

Reviewer #1:The authors set out to demonstrate the importance of a subset of cancer cells lacking CD44 and CD24, in promoting tumor progression, metastasis and resistance to therapy. They analyse archival breast tissues from patients where the clinical outcome – development of metastasis and therapy response – are known. They then take this information into cell culture experiments, seeking the mechanism that underlies the importance of these CD44-/CD24- cells. Their major premise is that these double negative cells can de-differentiate into cancer stem cells that subsequently promote metastasis and chemotherapy resistance.The overall concept of this study is good and the cell-based results are mainly clear. However, the archival tissue analysis needs clarification, as does the RNAseq analysis, as detailed below.

We thank the reviewer for his/her positive comments. We also recognize the limitations of our current study and have fully addressed the reviewers’ concerns by clarifying the analysis methods in the revision.

The breast cancer tissues are scored by immunofluorescence for expression of CD44 and CD24, but there is no indication that these two markers are being scored only in tumor cells rather than in all lineages within the tumor microenvironment. This is an important point that must be clarified. Also, the RNAseq analysis is reported to be conducted on single cells, but the data presented do not indicate if this is actually the case and how many replicate samples were analysed and which sample yielded each heat map pattern. Again clarification is required before the data can be interpreted.

We are sorry for the confusion. Actually, we analyzed the frequency of CD44-CD24- breast cancer cells in breast cancer tissue sections, but not stromal, infiltrates and other non-tumor cells, identified in the H&E stained corresponding series of breast cancer sections. Now, we have clarified it in the revision and provided the relevant H&E stained section in the supplementary Figure x. Please see our answers regarding the RNAseq method in question 12.

Finally, the conclusion from this study is that RHBDL2 induces spontaneous conversion of CD44-/CD24- cells into CD44+/CD24- cells via YAP1, USP31 and decreased NFkB signaling. This may well be the case, but has not been proven by the experiments presented here. Whilst DHBDL2 does alter YAP, USP31 and NFkB, it could also be working through an alternate pathway to induce CSCs.

We fully understand his/her concern. It is true that RHBDL2 acts as an intramembrane serine protease that can cleave and activate other substrates, including ephrin B and others, contributing to the development and progression of malignant tumors. Actually, ephrin B has been shown to function as a prognostic factor of uterine cervical (PMID
25205602 ). Unfortunately, we are unable to perform more experiments to test alternative mechanisms underlying the action of RHBDL2 in promoting distant metastasis of breast cancer due to the limited resources. In response to his/her concern, we have modified the conclusion of this study in a conservative manner and briefly discussed the potential alternative mechanisms underlying the action of RHBDL2.

1. Figure 1 and S1 and S2A score the numbers of samples with the varying CD44/CD24 protein levels. For Figure S1, this is shown by immunofluorescence, but there is no indication of whether the cells being analysed are tumor cells or host cells from the microenvironment. There should be co-staining for EpCAM or CK19 to confirm that tumor cells only are being scored. I am assuming that the data shown in Figure S2A are actually from the immunostaining and not from flow cytometry analysis as stated in the figure legend? If this was a flow analysis, there is no description of the origin or processing of the samples reported in Figure 1A for flow cytometry. The legend for Figure S2A states that the data are stratifying only metastatic tumors, but in Figure 1A, apparently many of the tumors showed no metastasis. All these points need to be clarified/corrected.

We fully understand his/her concern. In this study, we performed H&E staining and immunofluorescence simultaneously in a series of consecutive tissue sections. With H&E stained sections, we identified tumor cells and other types of non-tumor cells in the tumor regions. Subsequently, we analyzed the numbers of CD44-CD24- tumor cells in each section as described in the method section to determine the frequency of CD44-CD24- cells in total tumor cells. We have provided the relevant H&E stained sections in the revision and clarified the data in Figure S2A(now is Figure 1—figure supplement 2) by revising the figure legend in the revision. We are sorry for our carelessness to state wrong method in the Figure S2(now is Figure 1—figure supplement 2) legend

2. This first section of the manuscript is poorly presented and I am struggling to follow the arguments and data being provided. Many of the figures are mis-labelled in the text and they are not presented in consecutive order as discussed in the text. For some references to data in the text where the wrong figure has been cited, I cannot even find the correct figure.

We fully understand the reviewer’s concern and have corrected the figure citations accordingly.

3. Possibly the way the data are presented, with some figures referring to the training set and other figures to the test set is adding to the confusion when it is not clearly stated which set is under discussion at the time.

We appreciate his/her advice. Now, we have specified the figures and their descriptions in the revision, accordingly.

4. Late metastasis – better phrased as delayed metastasis.

We appreciate his/her advice. Now, we have changed the “late” into “delayed” metastasis in the revision.

5. The sentence in the Introduction: "Previous studies have shown that CD44+/CD24- breast CSCs might be a dominant factor in relapse of triple negative breast cancer (TNBC), due to their possession of potent self-renewal and differentiation capacities to differentiate into mature CD44-/CD24-, CD44+/CD24+, and CD44-/CD24+ cancer cells (Geng et al., 2014; Wang et al., 2014)". It is not at all clear to me that the references provided support this comment. Geng et al. do not mention CD44-/CD24- cells at all, and Wang et al. do not discuss differentiation of CD44+/CD24- into the other phenotypes. Only CD44- cells are mentioned, not CD24.

We thank the reviewer for his/her careful review. To avoid potential overstatement, we have modified the sentence into a general concept by deleting specific differential phenotypes in the revision.

6. Sentence in Introduction: "Indeed, injection of up to 1000 breast CSCs was able to generate a solid tumor mass in immunocompromised mice (Chaffer et al., 2011; Iliopoulos et al., 2011)." Please clarify. It is not obvious to me that Chaffer et al. specified how many CSCs they injected and Iliopoulos et al. showed that as few as 50 CSCs could form tumors.

We thank the reviewer for his/her careful review. We have re-checked the references and found that Chaffer et al. described that injection of up to 50 breast CSCs was able to generate a solid tumor mass in immunocompromised mice. Accordingly, we would like to keep the sentence as is.

7. Sentence in Introduction: "Moreover, previous studies reported that non-CSCs, such as CD44-/CD24- TNBC cells, are able to spontaneously convert into CSCs to renew the CSC pool, resulting in chemoresistance (Gruber et al., 2016; Kim et al., 2015; Ye et al., 2018)." The statement being referenced is for non-CSCs in TNBC, but Gruber et al. and Ye et al. are not reporting on TNBC.

We thank the reviewer for his/her careful review. Now, we have revised the text by changing “CD44-/CD24- TNBC cells” to “CD44-/CD24- breast cancer cells” to reflect the findings in the references.

8. Figure 1B – please explain more fully how the cut-off of 19.5% CD44-/CD24- was determined.

We thank the reviewer for his/her advice. the average frequency of CD44-/CD24- cancer cells in all samples was 19.7% and the median was 19.5%. The receiver-operating characteristic curve analysis also showed a decision threshold of 19.5% CD44^-^/CD24^-^ cancer cells; thus, we used this cut-off point to perform a subgroup analysis.

9. Figure 1C – please explain how the ratios of different subtypes of breast cancer with high or low CD44-/CD24- were calculated. According to the figure, for luminal cancers, there was a metastasis rate of ~40% in the high group and less than 10% in the low group. How were the numbers of 63.1% and 32.6% derived from Figure 1C?

Figure 1C describes the effect of CD44-/CD24- levels on the metastasis rates among three molecular subtypes of breast cancer, i.e., 63.79% vs. 22.22% (high vs. low CD44-/CD24- tumor cells in TNBC), 40.85% vs. 6.82% (luminal), and 47.06% vs. 13.64% (HER-2).

Then the same figures are provided for Figure 2A, when using the test set. Were the data in Figure 1C from the training set? Why are the final metastasis rates exactly the same? This needs a better explanation.

Figure 2A contains an error in the description and has been corrected to luminal: 22.73% vs. 8.89%; HER-2: 50% vs. 18.42%; TNBC: 41.67% vs. 15% for high vs. low CD44-/CD24- cells.

10. Page 6: reference to Figure S3C should be Figure S2C?

We thank the reviewer for their careful review and have corrected this figure citation.

11. Page 7 – Should Figure S5A,B actually be Figure S3?

We thank the reviewer for their careful review and have corrected the figure citation of Figure S5A,B to Figure S4A,B(now is Figure 2—figure supplement 1).

12. Page 7: The statement: "Moreover, a low CSC percentage led to different risks for developing tumor metastasis 5 years after diagnosis and adjuvant chemotherapy between the C1 and C0 patients (Figure S3D)." This does not appear to apply to Figure S3D and I cannot tell to which figure this statement is referring.

We are sorry for the confusion. The correct description should be that we stratified patients with <2% CD44+/CD24- and >19.5% CD44-/CD24- tumor cells in the C1 group while those with <2% CD44+/CD24- and <19.5% CD44-/CD24- tumor cells in the C0 group. We have clarified it in the text and figure legend accordingly.

13. Page 7, last sentence: "In the present study, we first designed the experiments illustrated in Figure 3A to characterize the percentages of different cell subtypes among MDA-MB-231 cells using cell culture and flow cytometric cell sorting of parental MDA-MB-468 cells." If I understand correctly, this should re-worded as follows: "In the present study, we first designed the experiments illustrated in Figure 3A and Figure S5 to characterize the percentages of different cell subtypes among MDA-MB-231 and MDA-MB-468 cells using cell culture and flow cytometric cell sorting of parental cells.

We thank the reviewer for his/her advice. Now, we have revised it in the manuscript accordingly. The data in Figure 3A, B have been updated with data of primary cells from breast cancer patients.

14. Figure 3C raises questions about reproducibility of the data. The profile in Figure 3C after 7 days in culture looks very different from that shown in Figure 3B.

We are sorry for the confusion. Actually, the Figure 3C shows the flow cytometric data of primary tumor cell-derived tumor in nude mice, which is different from the flow cytometric chart of the cultured cell lines. We have specified it in the Figure 3C legend.

15. Figure 5C: please explain how you obtained a weight measurement for the lung metastases. You record lung metastasis weights of up to 2 grams. Lungs typically weight about 0.2 grams. Lungs of 2 grams seem improbable, let alone metastases in lungs weighing 2 grams. Also, tumor sizes of 700 mm2 do not seem possible in lungs.

We are sorry for our carelessness. Actually, we measured the lung weights, but not the tumor weights because we cannot dissect all metastatic tumor nodules in the lung. Similarly, the tumor size data were calculated wrongly, based on the magnified images. Now, we have corrected these mistakes in the revision. Overall, our data indicated the converted CSCs had similar tumorigenicity and differential capacity in mice.

16. The RNAseq analysis requires more explanation. In Figure 6B, please explain the three columns shown in the heat map. Without their designation, the rest of this figure is hard to interpret. It seems that only one single cell per timepoint has been analysed. Is this correct? How reliable is sequencing from a single cell? Or was the analysis from many cells grown up from a single cell? How many single cells (or derivative of single cells) were analysed at each time point? There is a discrepancy between the protocol described in the text and that provided in the Methods section, when describing the RNAseq analysis. It is difficult to relate the significance of the genes shown in Panel B to the Kaplan-Meier data when it is not clear what the three columns in panel B represent.

We are sorry that we did not describe the experimental protocol in detail and present the data in Figure 6B well. We cultured CD44-/CD24- MDA-MB-231 cells in a single cell manner for 24, 72 and 120 h and they were harvested for RNAseq analysis of three single cells at each time point. We have modified the experimental protocol to specify it in the revision. Although we did single cell RNAseq the data could not be explained by single cell RNAseq, rather than general RNAseq. Accordingly, we changed the term of single cell RNAseq into RNAseq to avoid potential misleading in the revision. From the data in the heatmap, the repeated single samples display similar levels of each gene, indicative of its reliable nature. We are sorry for our carelessness and now, we have modified the manuscript to ensure the consistence between the protocol and result sections. In addition, we have labeled the columns in Figure 6B to specific the samples and time points.

17. Figure 6D: I assume that the "*" and "#" are meant to convey some sort of significance, but they do not appear for any of the genes assessed by RT-PCR. Does that mean that none of the changes were significant?

We are sorry for the confusion. To clarify the significance among groups of cells, we have specified the significance as * or # vs. CD44-/CD24- cells in the Figure legend. There were significant difference in the levels of mRNA transcripts of some genes between these types of cells in our experimental system.

18. Why does the legend for Figure S6 talk about SILAC for protein analysis? Was SILAC run on these samples? Impossible on single cells. No mention of SILAC in Methods.

We are sorry for our mistake. We did not run SILAC in this study. We have deleted it in the revision. (now is Figure 6—figure supplement 1)

19. Page 10: Explain why YAP was selected for analysis following knockdown of RHBDL2. Was there a previously known connection between these two genes? For the CD44-/CD24- cells in Figure 7B, was YAP phosphorylation also altered? Please show a western demonstrating that RHBDL2 is reduced after transfection with siRHBDL2.

We thank the reviewer for his/her valuable comments. Indeed, the Hippo pathway is important for the stemness of tumor cells. Thus, we selected the YAP1 as a verification indicator. We did not find any available data on how RHBDL2 modulates YAP1 expression and phosphorylation in any type of tumors in the literature. In response to his/her concerns, we have performed additional experiments and found that transfection with RHDBL2-specific siRNAA effectively decreased the relative levels of its expression in CD44-/CD24- cells. More importantly, RHDBL2 silencing also decreased YAP1 expression, but increased YAP1 phosphorylation (Figure 7B), leading to increased USP31 expression and decreased NF-κB phosphorylation in CD44-/CD24- breast cancer cells. These data supported our conclusion.

20. Figure 7D: text says 0.7% CD44+/CD24- but figure says 0.5% and vice versa for the MDA-MB-468 cells.

We are sorry for the confusion. Actually, the Figure 7D describes the data from MDA-MB-231 cells, which were 0.5% of CD44+/CD24- cells while the Figure 7E displays the data (0.7% of CD44+/CD24- cells) from MDA-MB-468 cells following RHDBL2 silencing. Now,we have clarified the data in the revision.

21. The text on page 11 needs re-writing. The second paragraph, commencing with "We next selected…" is a repeat of the text above where you have already done this step and analysed the cells by flow cytometry.

We thank the reviewer for their careful review and have revised the paragraph accordingly.

Reviewer #2:The manuscript describes a possible link between the presence of CD44-/CD24- cells and post-operative site distant metastasis. This is not a new concept, yet the actual biological function of CSC, particularly of the phenotype CD44-/CD24- remains inconclusive. The primary strength of the study is the large number of patient samples used in the retrospective analysis that provides the rationale for in vitro experiments using cultured cancer cells. Several technical and methodological limitations were identified.The description of the patient cohort is very elusive. The bioinformatic analysis lacks in rigor as the methodology used to identify and quantitate CD44-/CD24- cells in patients is not properly described. For instance, Figure legends describe the use of flow cytometry for quantification of CD44-/CD24- negative cells from patient samples. Yet only an immunofluorescence staining is described in Results. No Methods description is given to FACS of patient samples. The manuscript seems focused on TNBC, yet the reader has a difficult time understanding how many samples are actually from TNBC patients. Two TNBC established cell lines are mentioned in the manuscript but no rationale is provided as to why some experiments are performed only with MDA-MB-468 and some with MDA-MB-231. The same is true for several other experiments. The extensive molecular description of CD44-/CD24- cell dedifferentiation into CSCs deserves distinction but will need to be confirmed in patient samples before conclusions can be drawn with regards to the biological significance of these findings to breast cancer patients and the medical community at large.

We fully understand the reviewer’s concerns and we are sorry for the confusion in the manuscript. As we addressed the similar concerns from the reviewer 1, we have clarified the experimental protocols and corrected couple mistakes in the revision. Actually, we used H&E staining to identify tumor cells in breast cancer tissue sections and employed immunofluorescence to quantify the number of CD44-/CD24- tumor cells in the consecutive sections to determine the frequency of CD44-/CD24- breast cancer cells in total tumor cells in individual tissue sections. Moreover, *Supplementary file 1A* (Table S1) shows the number of patients with three molecular classifications and their clinical information. In addition, our data have been verified with two cell lines MDA-MB-468 and MDA-MB-231, and these data have been added to the manuscript. In order to confirm our findings in patients’ samples, we utilized fresh specimens from breast cancer patients to isolate primary cells for data verification. The corresponding results have been added in Figure 3.

Clarify the methodology used to quantitate CD44-/CD24- in patient samples, as well as the bioinformatic analysis supporting the current hypothesis.

We appreciate his/her advice. As we stated above, we have clarified the methods for quantifying CD44-/CD24- breast cancer cells in tissue sections above and in the revision. In addition, we have modified the section of bioinformatics to clarify the strategies for anlaysis of data in the revision.

Provide a detailed analysis of clinical and pathologic information of the patient cohort, including histology.

We appreciate his/her advice. Now, we have provided the detailed information on demographic and clinical characteristics of all patients we studied in *Supplementary file 1A* (Table S1) of the revision.

Several references are 10 years old or older, new references supporting the authors finds would strength the hypothesis.

We appreciate his/her advice. Now, have updated the reference section with new references available in the literature.